# A lexical approach for identifying behavioural action sequences

**Gautam Reddy** [1]*, **Laura Desban** [2], **Hidenori Tanaka**[3,4], **Julian Roussel** [2],
**Olivier Mirat**[2], **Claire Wyart** [2]*

**1** NSF-Simons Center for Mathematical & Statistical Analysis of Biology, Harvard University, Cambridge,
Massachusetts, United States of America, **2** Sorbonne Université, Institut du Cerveau (ICM), Inserm U 1127,
CNRS UMR 7225, Paris, France, **3** Physics & Informatics Laboratories, NTT Research, Inc., East Palo Alto,
California, United States of America, **4** Department of Applied Physics, Stanford University, Stanford,
California, United States of America

* gautam_nallamala@fas.harvard.edu (GR); claire.wyart@icm-institute.org (CW)

**Data Availability Statement:** The code required to reproduce the figures is available at https://github.com/greddy992/BASS. The zebrafish larvae data used to reproduce the figures is held at https://doi.org/10.5061/dryad.6t1g1jwwz.

## Abstract

Animals display characteristic behavioural patterns when performing a task, such as the spiraling of a soaring bird or the surge-and-cast of a male moth searching for a female. Identifying such recurring sequences occurring rarely in noisy behavioural data is key to understanding the behavioural response to a distributed stimulus in unrestrained animals. Existing models seek to describe the dynamics of behaviour or segment individual locomotor episodes rather than to identify the rare and transient sequences of locomotor episodes that make up the behavioural response. To fill this gap, we develop a lexical, hierarchical model of behaviour. We designed an unsupervised algorithm called "BASS" to efficiently identify and segment recurring behavioural action sequences transiently occurring in long behavioural recordings. When applied to navigating larval zebrafish, BASS extracts a dictionary of remarkably long, non-Markovian sequences consisting of repeats and mixtures of slow forward and turn bouts. Applied to a novel chemotaxis assay, BASS uncovers chemotactic strategies deployed by zebrafish to avoid aversive cues consisting of sequences of fast large-angle turns and burst swims. In a simulated dataset of soaring gliders climbing thermals, BASS finds the spiraling patterns characteristic of soaring behaviour. In both cases, BASS succeeds in identifying rare action sequences in the behaviour deployed by freely moving animals. BASS can be easily incorporated into the pipelines of existing behavioural analyses across diverse species, and even more broadly used as a generic algorithm for pattern recognition in low-dimensional sequential data.

## Author summary

Animals in the wild perform characteristic motor sequences during a task, for example, the surge-and-cast of a male moth while it searches for a female or that of a soaring bird spiraling up a thermal. Such sequences recur yet occur transiently and are not easily inferred from behavioural data. How can we find recurring yet transient action sequences from noisy behavioural data without access to stimulus information? To address this

**Funding:** This research was initiated during the summer school "Neural computation for sensory navigation" held in 2018 in the Kavli Institute of Theoretical Physics (University of California in Santa Barbara, USA) and was supported in part by the National Science Foundation under Grant No. NSF PHY-1748958, NIH Grant No. R25GM067110, and the Gordon and Betty Moore Foundation Grant No. 2919.01. This work was also supported by a New York Stem Cell Foundation (NYSCF) Robertson Award 2016 Grant #NYSCF-R-NI39, the HFSP Program Grants #RGP0063/2018, the Fondation Schlumberger pour l'Education et la Recherche (FSER/2017) for C.W. The research leading to these results has received funding from the program "Investissements d'avenir" ANR-10-IAIHU-06 (Big Brain Theory ICM Program) and ANR-11-INBS-0011–NeurATRIS: Translational Research Infrastructure for Biotherapies in Neurosciences. C.W. received support from the ERC-Proof Of Concept grant ERC-POC-2018#825273 for the development of the tracking algorithm ZebraZoom implemented by O.M. (www.zebrazoom.org). G.R. was partly supported by the NSF-Simons Center for Mathematical & Statistical Analysis of Biology at Harvard (award number #1764269). The funders had no role in study design, data collection and analysis, decision to publish, or preparation of the manuscript.

**Competing interests:** I have read the journal's policy and the authors of this manuscript have the following competing interests: C.W. and O.M. disclose a potential conflict of interest with the commercialization of the tracking algorithm ZebraZoom (www.zebrazoom.org).

question, we developed an unsupervised algorithm to extract an animal's action sequence repertoire in a manner analogous to how young children learn language from speech. Applying this approach on larval zebrafish, we uncovered a sequence of fast large-angle turns and burst swims that fish use to escape from an aversive environment. On simulations of a soaring bird, we recovered the characteristic spiraling patterns executed by the bird during thermalling. The algorithm can be more generally used to find rare but stereotypical patterns in low-dimensional sequential data.

## Introduction

Animal behaviour is extremely diverse and context-dependent. Yet, animals often exhibit certain characteristic patterns in their behaviour, particularly when executing a task or in response to a stimulus. One increasingly common computational approach to quantitatively describe behaviour is to leverage recent developments in the automated tracking of postural dynamics [1–8]. These methods exploit clusters in low-dimensional embeddings of postural dynamics to describe behaviour as a sequence of stereotyped elementary locomotor episodes (or bouts) drawn from a probabilistic model. The resulting descriptions parallel language models, containing information about local dynamics in the form of a probabilistic syntax over individual locomotor episodes (the classic example being a Markov model) [9–21].

An alternate, overlapping viewpoint is to interpret individual postures as segments of longer recurring action sequences or behavioural motifs, resulting in a description of behaviour as a sequence of 'words' with much less attention paid to transitions between characters. Motifs and transitions between motifs arguably contain more meaning in the context of a behavioural algorithm, much like in language. Indeed, in his seminal paper, Lashley [22, 23] rejects the *reflex chain* theory, which posits, in current terminology, a Markov model for the dynamics of movements in favor of a model based on noisy motifs.

While many methods exist to learn the dynamics of an animal's behaviour or cluster individual elementary locomotor episodes (see references above), few methods exist to efficiently infer recurring action *sequences* within a generative, probabilistic modeling framework. In this work, we aim to fill this gap and develop methods to identify recurring sequences of behaviour. This approach is particularly well-suited for highlighting the differences in navigation across different environments or genetic variants, which reveal how the animal reacts to an environmental or genetic perturbation. Such differences are often quite subtle, making comparative analyses difficult to implement using existing dynamical models. The key difficulty lies in that the majority of behaviourally-relevant responses are transient and occur only a few times in the dataset. By focusing on capturing short time-scale dynamics, dynamical models miss low-copy-number, behaviourally-relevant patterns. Such stretches are lost in the noise and are difficult to pick out from a large dataset.

To give a simple example, consider a scenario where one is presented with a control 'behavioural' dataset consisting of a sequence of 100,000 fair coin tosses, and a treatment dataset which is otherwise statistically identical except for 50 sequences of 20 consecutive tails placed at random locations within the sequence. By eye, the sequences in the treatment dataset clearly stand out as abnormal, relevant stretches. On the other hand, a Markov model on heads and tails, for example, when fit to the treatment data may indeed show a statistically significant deviation in its transition matrix from the control, but does not point to the nature of the abnormal stretches or where to find them. Similarly, during navigation in a spatially-distributed sensory landscape, robust responses to gradients of attractive or aversive chemical [24],

updraft [25, 26] or visual [27, 28] gradients are subtle. The moment at which the animal experiences and reacts to the cue will vary from one animal to the other and is difficult to infer. To identify repetitive elements of responses, one then has to find transient behavioural patterns that are over-represented in the environment enriched with chemical, thermal or visual gradients relative to a uniform control. To discover and construct dictionaries of recurring action sequences, referred to as 'motifs' [3], we developed a lexical model of behaviour and the unsupervised method BASS (Behavioural Action Sequence Segmentation). Motifs correspond to recurring sequences of elementary locomotor episodes of arbitrary length. A well-established motif for larval zebrafish corresponds to the sequence of locomotor episodes consisting of a J-turn followed by pursuit and capture swims deployed for hunting prey [17, 29, 34]. BASS has the following advantages: 1) BASS does not need to know the sensory information perceived by the animal, 2) BASS captures extended responses that last much longer than a typical locomotor episode, 3) BASS identifies recurring sequences despite them rarely occurring in the behavioural recording. Our key generative assumption is that animals navigating in a particular environment draw their behaviour sequentially from a context-dependent dictionary of motifs. Our goal then is to develop efficient, unsupervised methods to infer this dictionary based on noisy instantiations of the motifs observed in experiments.

If elementary locomotor episodes are represented by symbols, one straightforward approach to motif discovery is to simply enumerate over-represented sequences of $n$ symbols ($n$-grams) [30–33]. While the simplicity of $n$-gram models is attractive, the memory and computation time required increases exponentially with $n$ and dependencies longer than $n$ symbols are not captured. An alternative, more efficient approach is to maintain a set of possible sequences (in the form of a dictionary [35] or a suffix tree [36–39]) and add a new motif $m_1\, m_2$ to this set by concatenating two existing motifs $m_1$ and $m_2$ only if they are juxtaposed more often than chance. While the latter class of methods have enjoyed great success in bioinformatics [35, 37] and text processing [36, 39], the complexity of behavioural data poses an additional challenge. Data in bioinformatics and text processing consist of a well-defined sequence of letters (AGTC or the English alphabet respectively) with little variability in instantiations of a particular word (words are rarely misspelled). We identify three sources of variability in behavioural data: (1) Action pattern noise, which is the variability in instantiations of a particular motif template, (2) Locomotor episode noise, i.e., the variations in observed output, which may lead to a movement appearing as a similar one, and (3) Background variability due to rare and/or erratic movements. To make an analogy with speech learning [40], our task is similar to learning new words from *spoken* language (with no distinctive pauses separating the words) and given prior knowledge of phonology. Action pattern noise, in this analogy, corresponds (not exclusively) to stutters in speech, locomotor episode noise to substitutions of similar phonemes (for example, the aspirated /pʰ/ and the unaspirated /p/), and background noise to the utterance of unique proper nouns or unusual sounds. To take into account these sources of variability, we generalize a modeling framework from bioinformatics [35] by introducing an additional two-level hierarchical model. The lower level maps observed behavioural data to a latent state space (similar to going from a Markov model to a Hidden Markov model) and the second level introduces a model for noisy instantiations of motifs (Fig 1a). Despite the model's complexity, we show that inference is tractable and that motifs can be efficiently extracted from datasets of sizes within reach of current experiments. We discuss the relationship of BASS with the $n$-gram model and other models in more detail in the "Relationship with other methods and extensions" section.

The larval zebrafish is an interesting vertebrate model organism to investigate the emergence of behavioural action sequences and how they are used to navigate in chemical gradients. In order to survive, five days old zebrafish larvae actively explore their environment avoiding toxic cues and searching for food using stereotypical locomotor episodes consisting of bouts of

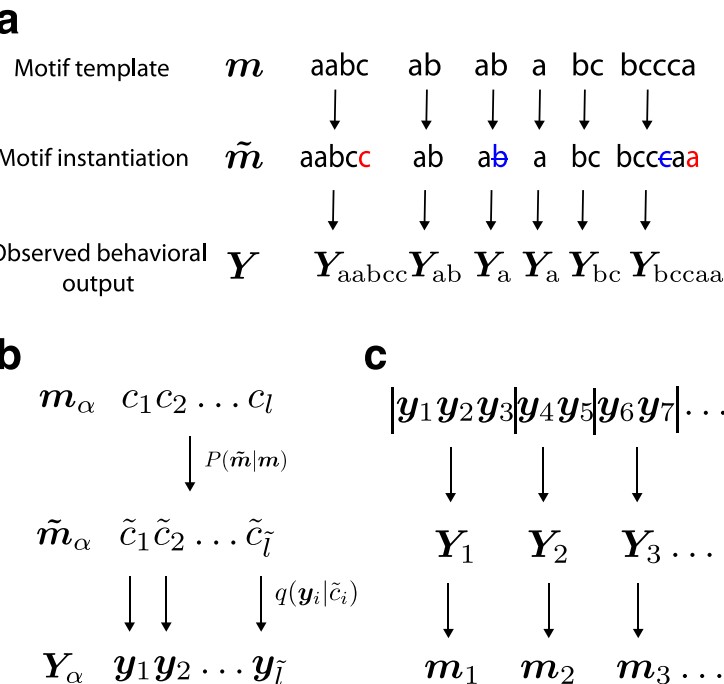

**Fig 1. The generative model from motifs to behavioural output.** (a) Motif templates are fixed sequences of elementary locomotor episodes (labeled a, b and c in this example). The observed behavioural output is generated from motif templates drawn sequentially from a dictionary. An instantiation of a template may "mutate" by insertions (red) or deletions (blue), which then generates the observed output as shown in panel (b). (b) The generative process from a motif template $c_1 c_2 \ldots c_l$ to instantiation $\tilde{c}_1 \tilde{c}_2 \ldots \tilde{c}_{\tilde{l}}$ to observed output $y_1 y_2 \ldots y_{\tilde{l}}$. (c) The unsupervised inference procedure (BASS) first learns a dictionary of motifs and then segments (vertical bars) the observed behavioural output $y_1, y_2, \ldots$ into the most likely sequence of motifs $m_1, m_2, \ldots$ from the dictionary that generated it.

activity lasting few hundreds of milliseconds separated by distinct pauses [16, 17, 29, 41]. Their small size enables the recording of numerous larvae in parallel, leading to the collection of thousands of swim bouts in a few minutes. Using our lexical approach, we first investigate the behavioural action sequences, i.e., the stereotyped sequences of bout types that larval zebrafish use to spontaneously navigate their environment. Next, we take advantage of a novel chemotaxis assay in which larvae navigate in arenas with gradients of noxious stimuli (acidic pH) and effectively avoid aversive regions. The behavioural response that enables zebrafish larvae to avoid aversive environments is unknown. Examining global kinematic parameters reveals only minor differences, which makes identifying the chemotactic response challenging with classical approaches and thus makes for an appropriate benchmark for our approach.

We first develop the lexical model and the motif identification algorithm, BASS. We apply the algorithm to synthetic data and to datasets obtained from freely-exploring and chemotactic zebrafish larvae. By comparing the dictionaries in the two environments, we identify the sequences that larvae use to chemotax. Lastly, we apply BASS to synthetic datasets of a soaring glider executing a thermalling strategy and show that the algorithm successfully identifies characteristic spiralling patterns.

## Model

### A lexical model of animal behaviour

Much like language, we assume the behaviour of an animal in a particular environment can be described by a sequence of motifs drawn from a dictionary $\mathcal{D}$, where each motif is a string of

arbitrary length containing characters from an alphabet. Motifs are to be considered as *templates* for the generation of action sequences. Each of the $K$ characters in the alphabet represent elementary locomotor episodes and correspond to the unique label of one of the $K$ soft clusters that define the elementary locomotor episodes in postural space. The probability density $q(\boldsymbol{y}|c)$, in practice obtained through clustering, specifies the probability of observing $\boldsymbol{y}$ when the animal executes the locomotor episode $c$. The implicit assumption here is the existence of well-defined elementary locomotor episodes, which has indeed been shown in a variety of systems including rodents, flies, worms and zebrafish larvae [11, 16, 41–43]. We may relax this assumption and instead use clustering as a tiling of postural space, which would manifest as additional noise and a larger alphabet.

Behaviour is generated from motif templates, which are sequentially sampled independently and identically from a distribution $\{p_{\boldsymbol{m}}\}$ over the motifs in the dictionary and individual characters (Fig 1a). The inclusion of individual characters accounts for movements that are not part of any motif, for instance, rare behaviours and erratic movements. These movements constitute background noise that impair motif identification since a motif $\boldsymbol{m} = c_1 c_2 \ldots c_l$ is detectable only if its likelihood is comparable to its constituent characters, $p_{\boldsymbol{m}} \gtrsim \prod_i p_{c_i}$. Given a sequence of motifs, the data is generated from each template $\boldsymbol{m}$ according to the probability density $Q(.|\boldsymbol{m})$ defined below, which is a central element of the model.

The probability, $Q(\boldsymbol{Y}_\alpha|\boldsymbol{m}_\alpha)$, of an observed output pattern $\boldsymbol{Y}_\alpha = \boldsymbol{y}_1 \boldsymbol{y}_2 \ldots \boldsymbol{y}_{\tilde{l}}$ given a motif template $\boldsymbol{m}_\alpha = c_1 c_2 \ldots c_l$ (Fig 1b) defines the behavioural output generated by $\boldsymbol{m}_\alpha$. We introduce a model for 'pattern noise': intuitively, if a motif template is viewed as the averaged trajectory of a stochastic dynamical system traversing through a state space, our model for pattern noise corresponds to one where in a particular realization, the trajectory spends a longer or shorter duration at certain regions of state space, but does not deviate into distant regions of state space. In particular, in each instantiation, $\boldsymbol{m}_\alpha$ 'mutates' to $\tilde{\boldsymbol{m}} = \tilde{c}_1 \tilde{c}_2 \ldots \tilde{c}_{\tilde{l}}$ with probability $P(\tilde{\boldsymbol{m}}|\boldsymbol{m}_\alpha)$. The output $\boldsymbol{y}_i$ is drawn independently for each character in the mutated sequence from $q(\boldsymbol{y}_i|\tilde{c}_i)$. To quantify pattern noise, we fix the probability of error per character that results either in the deletion or duplication of that symbol. Note that locomotor episode noise is implicitly incorporated via soft clustering and is determined by the discriminability of neighboring states. We derive a recursive equation for the efficient calculation of $Q(\boldsymbol{Y}_\alpha|\boldsymbol{m}_\alpha)$ (see Materials and methods).

Performing inference on this model requires constructing the dictionary $\mathcal{D}$ as well as estimating the motif probabilities $\{p_{\boldsymbol{m}}\}$. To build our dictionary, we use an iterative procedure generalized from ref. [35] to our latent space model, where we start from a dictionary with only single characters and progressively add new motifs based on how often existing motifs occur next to each other. In particular, we cycle between: (1) estimating $\{p_{\boldsymbol{m}}\}$ using maximum likelihood estimation (MLE), (2) expanding $\mathcal{D}$ if certain pairs of motifs occur next to each other more often than you would expect from $\{p_{\boldsymbol{m}}\}$, (3) truncate shorter motifs from $\mathcal{D}$ that are "explained away" by the addition of the longer motifs into the dictionary. We briefly expand on these three steps in the next paragraph; see Materials and methods for further details.

Given a behavioural dataset $\boldsymbol{Y} = \boldsymbol{y}_1 \boldsymbol{y}_2 \ldots \boldsymbol{y}_L$, the series of motif templates that generate it is unknown. For example, if $L = 3$, we have $\boldsymbol{Y} = \boldsymbol{y}_1 \boldsymbol{y}_2 \boldsymbol{y}_3$, whose likelihood is obtained by summing over all possible ways the dataset can be partitioned: $Q(\boldsymbol{y}_1)Q(\boldsymbol{y}_2)Q(\boldsymbol{y}_3) + Q(\boldsymbol{y}_1)Q(\boldsymbol{y}_2 \boldsymbol{y}_3) + Q(\boldsymbol{y}_1 \boldsymbol{y}_2)Q(\boldsymbol{y}_3) + Q(\boldsymbol{y}_1 \boldsymbol{y}_2 \boldsymbol{y}_3)$, where each marginal probability factor in each term is from an instantiation of a particular motif template. In general, the likelihood of $\boldsymbol{Y}$ under our generative model is the sum over all possible partitionings $\{\pi\}$ of the dataset (of which there are $2^{L-1}$)

into observed data sequences $\{Y_\alpha^\pi\}$, weighted by the likelihood of each partitioning:

$$P(Y; \{p_m\}) = \sum_\pi \prod_{\alpha=1}^{N(\pi)} Q(Y_\alpha^\pi), \tag{1}$$

where the marginal probability is $Q(Y_\alpha^\pi) = \sum_m Q(Y_\alpha^\pi|m)p_m$ and $N(\pi)$ is the total number of templates in partitioning $\pi$. We show (Materials and methods) that the MLE for $p_m$ satisfies the implicit equation

$$p_m^* \propto \sum_\pi \sum_{\alpha'=1}^{N(\pi)} p(m|Y_{\alpha'}^\pi) \prod_{\alpha=1}^{N(\pi)} Q(Y_\alpha^\pi), \tag{2}$$

where $p(m|Y_\alpha^\pi)$ is the posterior probability of $m$ given the data and the pre-factor is determined from normalization. The sum over the posterior probabilities can be interpreted as an effective number of counts of $m$ in the partitioning $\pi$; Eq (2) can then be re-cast as $p_m^* = \langle N_m \rangle / \bar{N}$, where $\langle N_m \rangle$ is the expected number of counts of $m$ over the ensemble of partitionings and $\bar{N} = \sum_{m'} \langle N_{m'} \rangle$ is the average number of partitions.

Given the implicit dependence of $p_m^*$ within the large sum in Eq (2), it is rather surprising that the MLE can be performed efficiently. To compute $p_m^*$, it is useful to define the free energy, $F \equiv -\ln P(Y; \{p_m\})$, which is to be minimized. The gradients of $F$ can be efficiently calculated using dynamic programming methods (Materials and methods), which allows for computation of $p_m^*$ using standard gradient descent methods. Note that the number of counts is then $\langle N_m \rangle = -p_m \partial_m F$. New motifs are added to the dictionary if they occur more often than expected by random concatenations of motifs already in the dictionary. The probability of a new motif $m$ being generated through all possible concatenations of smaller motifs in the dictionary, $\zeta(m)$, is compared to the empirical probability of $m$, $-\zeta(m)\partial_m F/\bar{N}$. A standard likelihood ratio test yields a $p$-value and pairs below a $p$ threshold ($10^{-3}$) are added to the dictionary. The results are not sensitive to this threshold. The output of the algorithm is a dictionary of motifs and the number of times each motif occurs in the dataset, $\langle N_m \rangle$. Standard statistical tests on $\langle N_m \rangle$ can then be used in order to compare the relative abundance or rarity of a motif across datasets (for example, a control and a treatment dataset). Further, a Viterbi-like algorithm can be used to partition the data into the most likely partitioning (Materials and methods). An implementation of BASS is publicly available [44]. Data deposited in the Dryad repository [45] www.doi.org/10.5061/dryad.6t1g1jwwz.

## Results

### An illustration on synthetic data

To illustrate the generative process and the effectiveness of the method in identifying and segmenting motifs, we first apply it to a synthetically generated dataset. We assume individual data points are two-dimensional (representing a lower-dimensional embedding of postural dynamics) and are drawn from 7 distinct states (which make up the characters in our alphabet) with a Gaussian emission function as shown in Fig 2a. A dictionary of 50 motifs is constructed such that each motif has a mean length of five. Given the generated dictionary, the probability of each motif, $p_m$, is drawn and scaled with a parameter $1 - \epsilon_b$, where $\epsilon_b$ is the fraction of the dataset that is made up of individual characters. We use $\epsilon_b$ as a measure of 'background noise'. Sequential data is sampled according to the lexical model, with $\epsilon_p$ as a measure of action pattern noise and locomotor episode noise $\mu$, defined as the distance between neighboring clusters

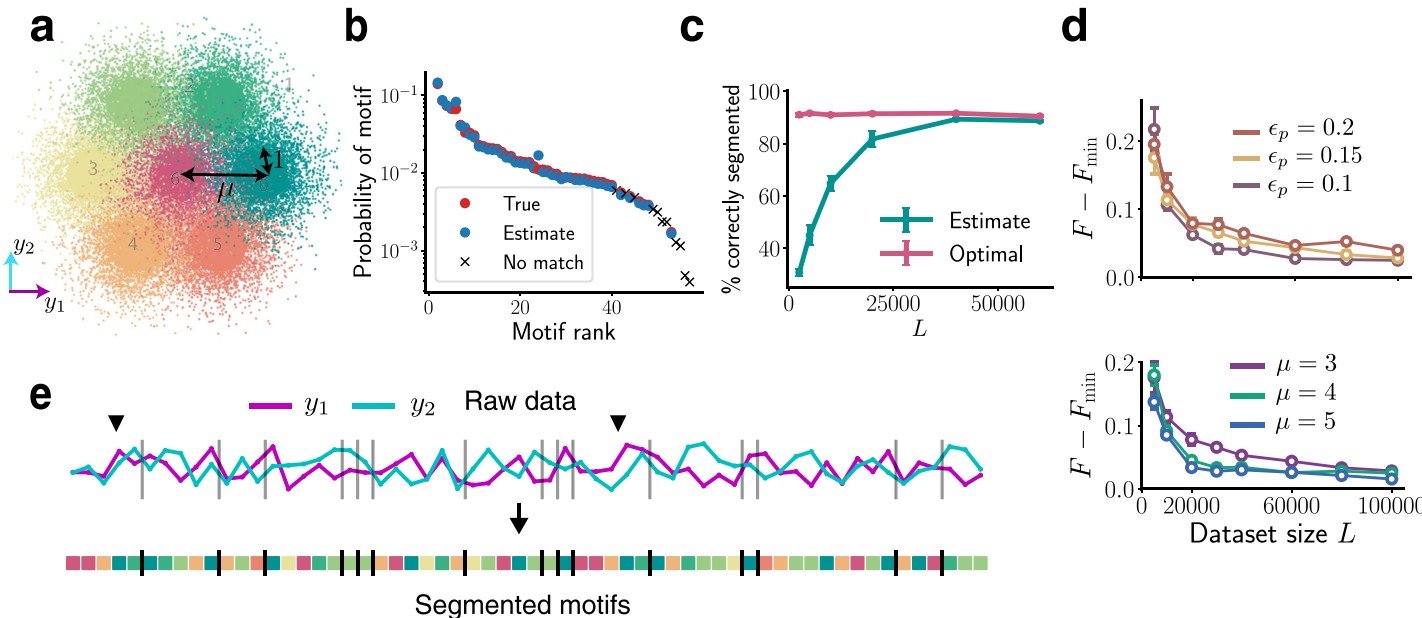

**Fig 2.** BASS accurately identifies and segments motifs in noisy, synthetic data: (a) The seven clusters from which the two-dimensional data (along $y_1$, $y_2$) is drawn. (b) The true probabilities of the motifs (red dots) and probabilities estimated (blue dots) by our algorithm showing successful reconstruction of the dictionary. The crosses are low-probability motifs not identified by the algorithm (see main text). (c) The percentage of correct segmentations into motifs (cyan) with increasing dataset size. The optimal percentage when the true dictionary is known is shown in pink. In panels b,c and e, we use $L = 40000$, $\epsilon_p = 0$, $\epsilon_b = 0.5$, $\mu = 3$. See S1 Fig for the case $\epsilon_p > 0$. (d) The difference in the negative log-likelihood per symbol after convergence when the true dictionary is unknown ($F$) and known ($F_{\min}$). Action pattern noise $\epsilon_p$ and locomotor episode noise $\mu$ are successfully integrated out with larger datasets. Top: $\mu = 3$, $p_d = 0.5$, Bottom: $\epsilon_p = 0.15$, $p_d = 0.5$, where $p_d$ is the probability of a deletion in a motif instantiation. Error bars are s.e.m. (e) A snippet of the raw data sequence and the most likely partitioning into motifs found by the algorithm. The vertical bars delineate two successive motifs. The black arrows mark two instantiations of the same length-five motif.

relative to the standard deviation of each cluster (Fig 2a). In the sample shown, we use $L = 40000$, $\epsilon_p = 0$, $\epsilon_b = 0.5$, $\mu = 3$.

On this dataset, the algorithm builds a dictionary containing 44 motifs with 11 false negatives and 6 false positives. Of the 11 false negatives (crosses in Fig 2b), 8 occur fewer than 25 times in the entire dataset. The three other false negatives (632, 631, 421325 in the inferred dictionary, see Fig 2a for cluster labels) were in fact closely related to the three most prominent motifs that were not identified (32,31,421335 in the true dictionary). The estimated probabilities of the true positive motifs match very well with their true probabilities (Fig 2b) despite significant background and locomotor episode noise. As a measure of performance, we compute the percentage of the sequence correctly segmented (i.e., both the state and the partition between motifs are correctly identified) by BASS in the most likely partitioning of the sequence. Fig 2c shows near-optimal performance for larger datasets, where optimal is defined with respect to the case when the true dictionary is known. Note that optimal partitioning performance may not be perfect ($\sim$90% in Fig 2c) as motif generation is a stochastic process. A snippet of the raw data is shown in Fig 2e along with the most likely partitioning into motifs from the learned dictionary. With larger datasets, the method robustly integrates out fluctuations due to significant action pattern and locomotor episode noise $\epsilon_p$ and $\mu$ (Fig 2d). BASS found no motifs in shuffled data.

We now apply BASS to larval zebrafish behaviour in exploratory (pH neutral) and aversive (acidic) chemotaxis assays. An outline of our analysis pipeline is shown in Fig 3a.

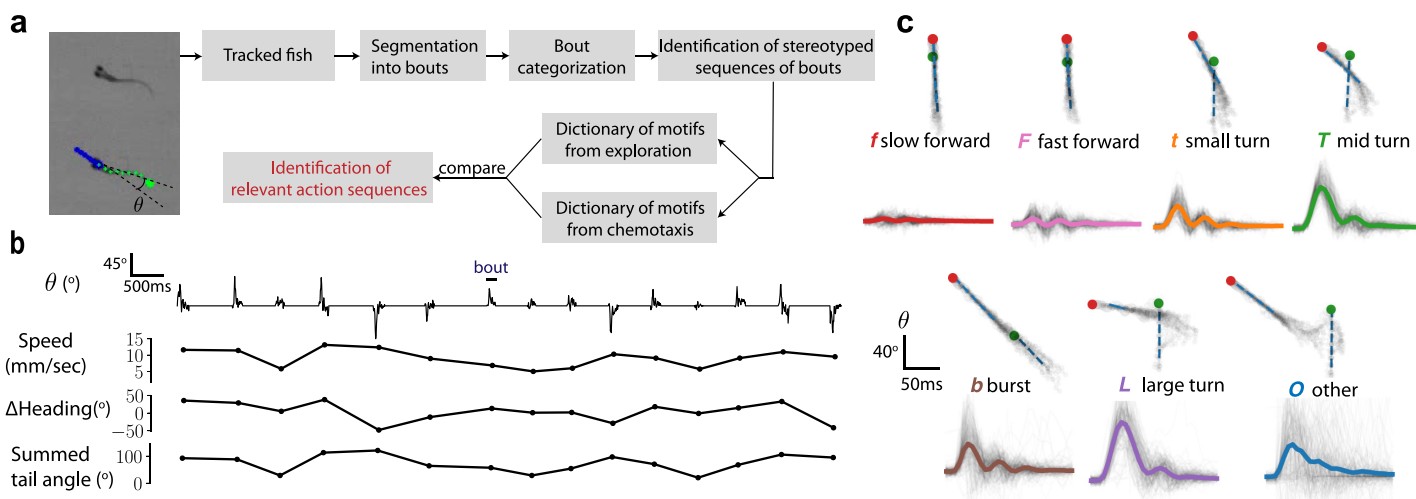

**Fig 3. Analysis of larval zebrafish behaviour exploring neutral and aversive environments.** (a) Overview of the analysis pipeline. (b) A time series of the tail angle $\theta$ shows the discrete nature of locomotor episodes ('bouts') with the corresponding speed, change in heading and the summed absolute amplitude of the tail angle calculated as the summed absolute amplitude of the tail angle. (c) Samples of the seven bout types identified using a Gaussian Mixture Model. In the superimposed images corresponding to a given bout type, the green and red dots correspond to the head position at the bout beginning and end respectively. Below each sample, the average tail angle $\theta$ is shown in solid color with 200 trajectories shown in grey.

## A dictionary of action sequences for exploring larval zebrafish

Zebrafish larvae swim in short punctuated locomotor episodes called 'bouts' (duration mean ± s.d = 150±50 ms) separated by longer periods of rest (mean ± s.d = 700±400 ms) (Fig 3b). Larvae spontaneously explore their environment by performing mainly slow bouts occurring as forward swims and routine turns, often by repeating turns in the same direction [46], and rarely exhibit fast bouts such as burst swims or escapes [16, 41]. We collected a dataset of ≈ 85000 bouts from 171 fish swimming in elongated swim arenas.

A single bout is well-characterized by the fish's tail movement and other kinematic variables such as average speed and change in heading. From raw tracking data (Materials and methods), we use a six-dimensional parameterization $y$ for each bout, which includes the speed, the change in heading, the summed tail angle (summed absolute amplitude of the tail angle) and the first three principal components of the tail angle over time (Materials and methods). Based on this parameterization, bouts were categorized into different bout types using a Gaussian Mixture Model (GMM). A GMM yields the probability density, $q(y|c)$, for each bout type $c$, which serves as a statistical description of each bout type in terms of the means and covariances of the six variables. We clustered bouts into seven bout types (Fig 3c and S1 and S2 Movies), which correspond to two forward swims of different speeds ($f$, slow and $F$, fast), three turns based on the magnitude of change in heading ($t,T$ and $L$, increasing angle), bursts ($b$) and an other ($O$) category. The $O$ category contained a variety of different bouts that did not clearly fall into one class; these included O-bends, long turns and bursts, and improperly tracked bouts. The bout types are not sharply delineated; this is not an issue for the BASS algorithm since variability in $y$ is implicitly taken into account via $q(y|c)$ as noted before.

Typical bout types are displayed in Fig 3c. Compared to previous categorizations performed on spontaneous exploration [16, 17, 29], our categories (except $O$) likely correspond to subdivisions of forward swims, routine turns and burst swims. The $f$, $t$, $T$ bout types typically correspond to the slow regime of locomotion enriched during basic exploration, while $F$, $b$ and the rare $O$ belong to the fast regime and occur overall less frequently when no aversive stimulus is applied [16, 41].

Zebrafish locomotor episodes in our conditions consist of seven bout types that make up the alphabet of our generative model. Sequences of consecutive bouts for each fish ($\approx 500$ bouts per fish) served as input to BASS. A coarse exploration of the pattern noise parameter $\epsilon_p$ and the probability of an insertion $p_d$ using a held-out dataset yielded $\epsilon_p = 0.1$ and $p_d = 0.2$, which were used for the rest of our analysis (S5 Fig). These numbers suggest noisy motif instantiation and a bias towards insertions (i.e., repeats). Notably, the greater held-out likelihood for $\epsilon_p > 0$ highlights the advantage of incorporating variability into our modeling framework.

The algorithm converged to a dictionary consisting of 66 motifs with similar results across trials and subsamples. The output of the algorithm is a dictionary consisting of the identified motifs and the expected number of occurrences of the motif, $\langle N_m \rangle$, in the dataset. Intuitively, $\langle N_m \rangle$ is the number of times a motif occurs after appropriately discounting its occurrences within a longer motif and taking into account pattern noise and locomotor episode noise. For example, a locomotor episode which is on the boundary between the *f* and *t* clusters (locomotor episode noise) is appropriately weighted as one half *f* and one half *t*. This example further highlights the importance of incorporating locomotor episode noise as a source of variability. Similarly, the observed sequence *ffff* will also contribute to the number of counts of the motif *fff* since the extra *f* could be due to an insertion (pattern noise). The observed sequence *ffff* will not contribute as one count for the motif *ffff* and also contribute as two counts for the motif *ff*. The counts are instead shared between the motifs *ffff* and *ff* based on the relative probability of the two motifs (implicitly using Bayes' rule) and the number of ways *ff* can appear in the observed sequence *ffff*. A subset of these motifs is shown in Table 1 (see also S1 Table). In Fig 4a and 4b, we provide a typical series of bouts segmented into sequences of motifs.

**Table 1. Motifs over-represented in the exploratory dataset.** A subset of motifs occur ('Observed' column) more often than predicted by a first-order Markov model (the 'Expected' column). The *p*-value is obtained using a likelihood ratio test. Single-length motifs are not shown (see Fig 5c). See also S1 Table. The fifth column shows the percentage of unique fish (out of 171 total) which executed the motif at least once in the most likely partitioning of the data. Note that a motif may never appear in the most likely partitioning even though it has non-zero $p_m$ (for eg. *tttt* below, which is partitioned into four individual *t*s). The rightmost column shows the fraction of the exploratory dataset tiled by each motif.

| Motifs | $-\log_{10} p$ | Observed | Expected | % of fish | Coverage |
|---|---|---|---|---|---|
| ffffffffff | >300 | 1366 | 387 | 51% (87/171) | 1.30% |
| ffffffffffff | >300 | 510 | 50 | 38% (65/171) | 1.67% |
| fffffff | 208.01 | 3234 | 1797 | 69% (118/171) | 2.01% |
| ffff | 42.33 | 9544 | 8327 | 93% (159/171) | 3.70% |
| FFFFFFF | 28.23 | 311 | 153 | 53% (91/171) | 1.47% |
| fffffftf | 27.64 | 497 | 290 | 70% (119/171) | 2.02% |
| fftfffff | 25.07 | 495 | 297 | 66% (113/171) | 2.11% |
| fftfff | 22.72 | 1125 | 824 | 76% (130/171) | 2.01% |
| fftff | 21.12 | 1745 | 1377 | 84% (144/171) | 2.78% |
| fftf | 18.5 | 2724 | 2289 | 57% (98/171) | 1.33% |
| ftf | 13.96 | 4337 | 3859 | 92% (158/171) | 4.10% |
| TfT | 11.12 | 722 | 554 | 68% (116/171) | 0.95% |
| FFFF | 7.94 | 1428 | 1224 | 67% (115/171) | 1.17% |
| TfTf | 7.28 | 346 | 254 | 58% (100/171) | 0.68% |
| tttt | 6.7 | 256 | 181 | 0% (0/171) | 0.38% |
| TTTT | 5.06 | 160 | 110 | 54% (92/171) | 0.41% |
| bb | 3.87 | 924 | 1044 | 30% (52/171) | 0.15% |
| bbbb | 3.21 | 115 | 82 | 21% (36/171) | 0.29% |
| FbFb | 2.19 | 99 | 74 | 19% (32/171) | 0.45% |

The dictionary reveals several surprising features. A significant fraction of the fish movements were made of motifs: motifs covered on average 78% of all bouts per fish with a standard deviation of 7.3% across fish. We found motifs as large as 14 bouts, and this could further expand in a particular realization due to insertions. In particular, *f* repeated 14 times occurred more than 500 times. While this may be explained by the large fraction of *f*, repeats were also found for *T*, *F* and *b*. Overall, the most enriched and common motifs correspond to repetitions of the same bout type, and typically occur 2–14 times in a row. Motifs containing mixtures of bouts included typically only 2 different bout types. Remarkably, we noticed throughout the list of enriched motifs that the two bout types belonging to a given motif, belonged either to low speed (mixtures of either *f* and *t* in *ffTf* or in *TfTf*) or to high speed (mixtures of *F* and *b* in *FbFb* or *bbFb*), but never combined fast and slow locomotor episodes.

To quantify how unusual these sequences were under a Markov model, we compared the observed occurrence of the identified motifs to those predicted from the best-fit Hidden Markov Model (HMM). Our lexical model yielded a better fit compared to an HMM (difference in held-out free energy per bout of 0.12), and a significant portion of motifs deviated from Markovianity (Table 1, S1 Table). Two aspects of the behaviour likely lead to the observed non-Markovianity. First, while long repeats of the same bout type occur often, the distribution of the number of repeats has a heavy tail and decays much slower than a geometric distribution (Table 1, S1 Table). Second, sequences with mixtures of two bout types such as *TfTf* and *fftf* are common; while the repeats emphasize (say) *f*→*f* transitions, the motifs with mixtures of bout types on the other hand emphasize *f*→*t* transitions, creating a tension between the two in a purely Markovian picture. One concern could be that the *t* bouts in the mixtures of *f* and *t* (e.g. *fftf*) lie on the border between the *f* and *t* clusters, and that the apparent non-Markovianity is simply due to excessive coarse-graining of the elementary locomotor episodes. However, this issue is solved by our soft clustering procedure, which assigns the appropriate probability weight to locomotor episodes that lie on the boundary between two clusters.

To verify that the long chain of repeats were not an artifact due to our elongated well geometry, we applied a similar pipeline of bout categorization and motif identification on a previously published dataset [16] (see Materials and methods, S4 Fig). The dataset consists of $\approx$ 120,000 bouts obtained from 23 fish freely swimming in a square well (of side $\sim$ 25mm) under varying light intensities. Notably, the resulting dictionary also displays long chains of repeats and significant non-Markovianity albeit with a heavier emphasis on turns compared to forward swims (S2 Table). We observe again in the motifs a dichotomy between the classical slow and fast locomotor episodes when bouts occur either via forward swims or turns. Mixtures are composed either of slow or of fast locomotor episodes: slow turns and forward swims appear together in a set of sequences (*ffTf*, *TfTf*) while fast forward and burst swims (*FbFb*, *bbFb*) appear together in other sequences. Remarkably, mixtures of *slow* bouts (forward swims or turns) and *fast* bouts (forward or burst swims) are conspicuously absent in both dictionaries.

## Fish chemotax away from acidic pH using action sequences of fast bursts and large avoidance turns

Although few studies have reported that zebrafish respond to acute applications of chemicals in the surrounding water [47–49], the behavioural responses of freely-swimming zebrafish larvae navigating in chemical gradients of aversive or appetitive cues have not yet been investigated. We applied acid to the two ends of our extended arenas (of length 14 cm) forming a sharp gradient (Materials and methods); diffusive transport at the time scale of the experiment (ten minutes) is at most 1 cm and therefore is confined to the ends. Zebrafish larvae robustly

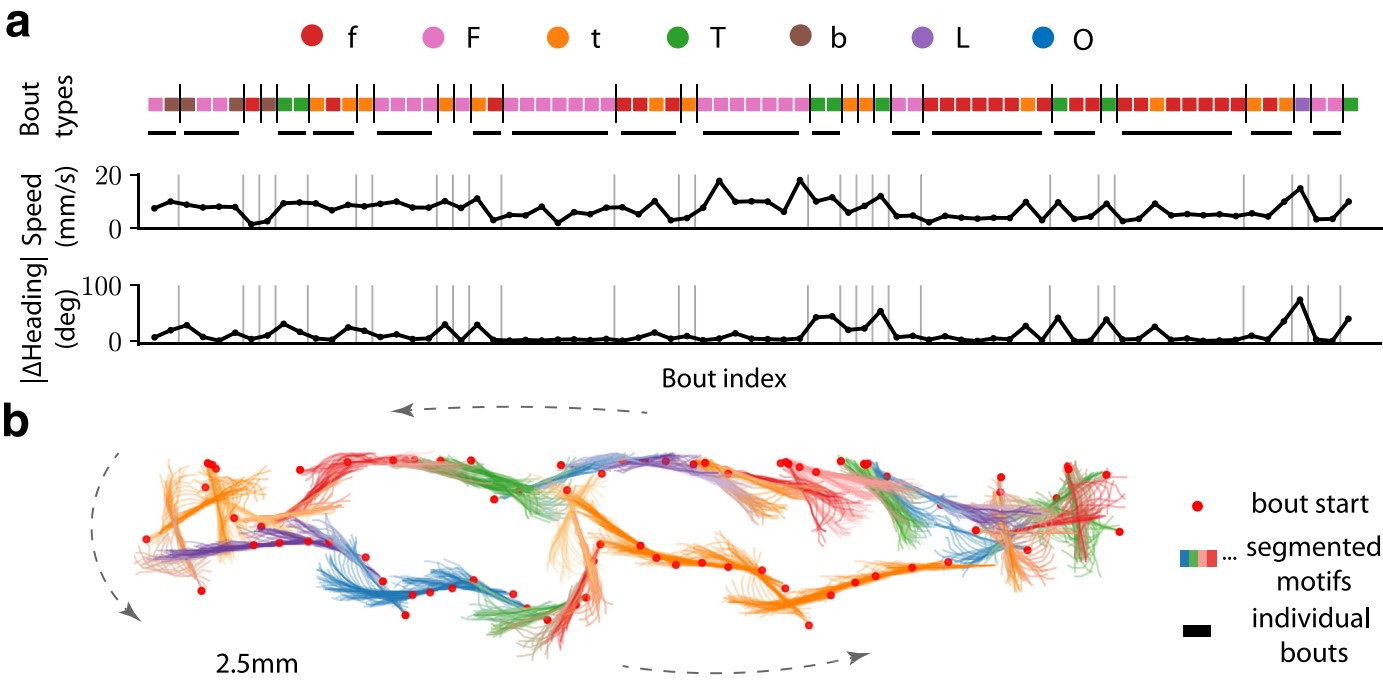

**Fig 4. Motifs identified by BASS make up a significant fraction of the dataset.** (a) A sample sequence of 75 bouts from the exploratory data segmented (separated by vertical bars) into the most likely sequence of motifs from the learned dictionary. The corresponding speed and absolute change in heading are shown. Motifs longer than one locomotor episode are underlined in gray. (b) A sample trajectory consisting of 80 bouts (head position at the beginning of a bout is shown as a red dot) are segmented into motifs (head and tail at each frame are shown), where successive bouts from the same motif have the same color. The black-colored segments of the trajectory are motifs of length one i.e., single locomotor episodes.

performed chemotaxis and avoided the two extremities (Fig 5a) despite displaying only minor differences in kinematic parameters (Fig 5b). The distribution of bout types was similar for larval zebrafish navigating in acidic gradients. We noticed a small over-representation of certain bout types (Fig 5c), suggesting that fish perform more fast forward swims *F*, burst swims *b*, and large-angle avoidance turns *O* in response to the aversive gradients. Burst *b* and fast forward swims *F* occurred uniformly around the center of the well (S3 Fig), while larval zebrafish executed more large-angle avoidance *O* bouts closer to the acidic gradient, suggesting that *O* bouts are potentially implicated in the response to noxious stimuli during aversive chemotaxis.

We next investigated whether the chemotaxis response extended beyond a single bout to action sequences composed of a sequence of bouts that were consistently observed across fish. We implemented a comparative approach of finding recurring sequences of actions that are highly over-represented in the aversive environment compared to exploration. We applied the BASS algorithm to the dataset from fish in the aversive environment ($\sim$66,000 bouts from 135 fish). The resulting dictionary of motifs contained a total of 81 motifs, slightly larger than the one obtained from exploration (S3 Table). The two dictionaries contain broad similarities: both contain long repeats of the same bout type and mixtures of *t* and *f*, yet contain important differences, particularly in the over-representation of mixtures of *b,F* and *O,b* in the aversive environment.

We examined over-represented sequences by comparing the relative occurrences of motifs in the exploratory and aversive environments. The two dictionaries were combined to obtain a total of 103 unique motifs and the expected number of occurrences of each motif, $\langle N_m \rangle$, for the two environments. We use $-\log_{10} p$ as a measure of over-representation, where $p$ is

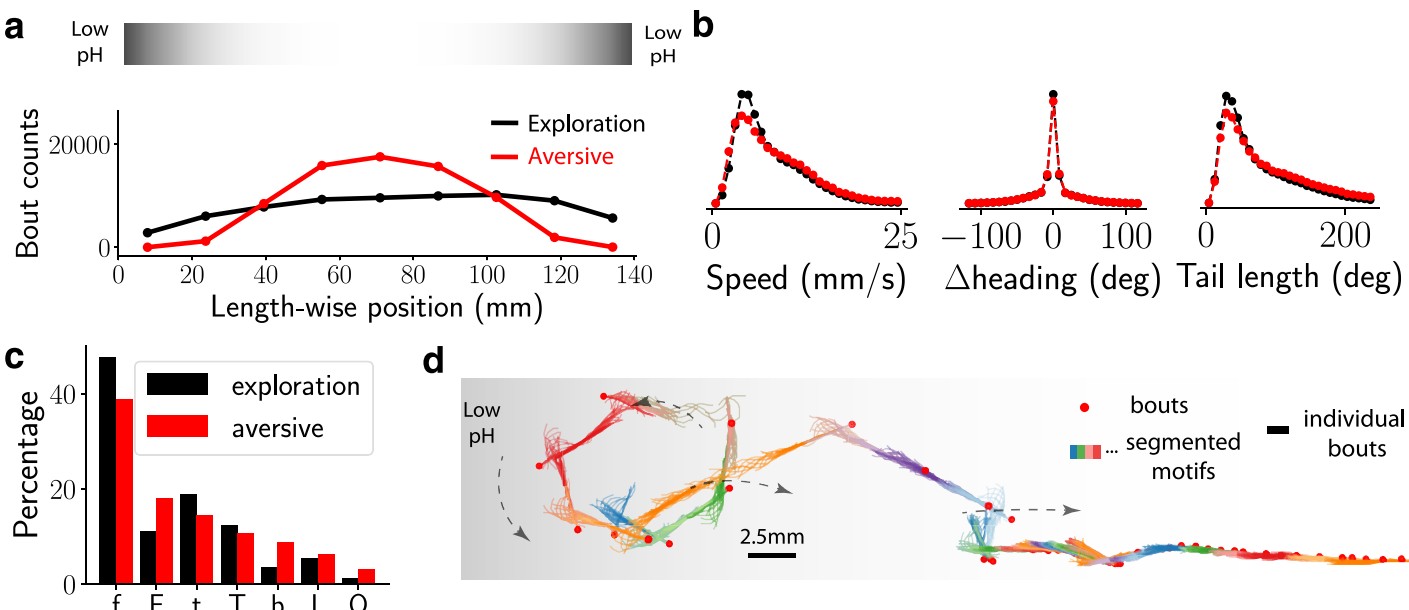

**Fig 5. Fish avoid aversive environments using a transient chemotactic response.** (a) Histogram of larvae positions along the well with and without the aversive (acidic) gradient, located at the ends of the well. The s.e.m is square-root of the counts, which is negligible. An illustration of the aversive gradient is shown above. (b) The distribution of speed, change in heading and the summed tail angle of all bouts during exploration (black) and in aversive environment (red). The difference in global kinematic parameters between the two environments is small. (c) The fraction of each bout type in exploratory and aversive environments, where a total of ≈ 85000 and ≈ 66000 bouts were collected respectively, shows an increase in fast bouts: *b*,*F* and *O*. (d) Localisation of bout types along the well with and without the aversive (acidic) gradient, (e) BASS segments a series of bouts from the aversive environment into sequence of recurring motifs. Shown here is a sample trajectory (as in Fig 4b) where the fish escapes from the aversive environment.

obtained from a likelihood ratio test on $\langle N_m \rangle$ in the two environments. To calibrate this score, we first split the exploratory dataset into halves and computed the $-\log_{10} p$ for each motif; the threshold 15 was chosen for a false positive rate of 10%. To find action sequences that were executed by many fish and not just a few abnormal fish, we sub-sampled our dataset from the aversive environment to 80% its size ten times, performed the comparison for each sub-sample, and chose only those motifs above threshold of 15 in *all* ten sub-samplings.

Table 2 shows the eight over-represented motifs found across fish. The eight motifs were executed by multiple fish (rightmost column of the table). All motifs except one (*fTff*) are mixtures of *b* and *O*. The bouts from the selected motifs (except *fTff*) are significantly over-represented close to the aversive gradient compared to the rest of the bouts in the aversive

**Table 2. Motifs consistently over-represented in the aversive chemotaxis assay.** The fifth column shows the percentage of unique fish (out of 135 total) which executed the motif at least once. The rightmost column shows the fraction of the aversive dataset tiled by each motif.

| Motifs | $-\log_{10} p$ | $\langle N_m \rangle_{\text{aver}}$ | $\langle N_m \rangle_{\text{explo}}$ | % of fish | Coverage |
|---|---|---|---|---|---|
| fTff | 90.84 | 101 | 5 | 64% (87/135) | 1.73% |
| OO | 66.63 | 111 | 11 | 60% (81/135) | 0.15% |
| bb | 55.84 | 210 | 55 | 82% (82/135) | 0.53% |
| bOOb | 46.06 | 63 | 5 | 46% (46/135) | 0.45% |
| Ob | 35.74 | 135 | 36 | 77% (77/135) | 0.56% |
| bO | 34.6 | 140 | 39 | 82% (82/135) | 0.53% |
| bFbb | 24.96 | 49 | 6 | 49% (49/135) | 0.54% |
| Obbb | 24.07 | 56 | 9 | 49% (49/135) | 0.52% |

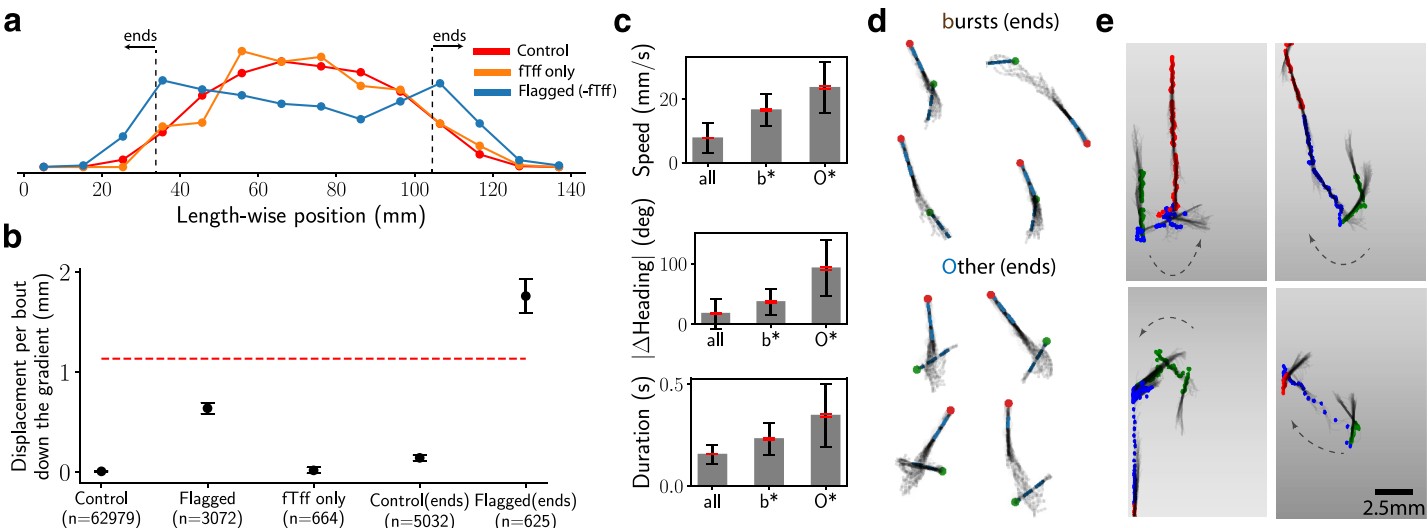

**Fig 6. Fish chemotax against an acidic gradient via sequences of fast bursts and large-angle avoidance turns.** (a) Distributions of length-wise positions for the Control bouts (all bouts from aversive environment except the ones from sequences flagged as over-represented in Table 2), *fTff* only and Flagged bouts (from the sequences in Table 2) except *fTff* in red, orange and blue respectively. (b) The length-wise displacement travelled in a bout down the gradient for bouts tagged as Control, Flagged (as defined in (a)), *fTff* only, Control(ends) (all bouts from the two ends of well shown in (a)), Flagged(ends) (flagged but with *fTff* removed and in the ends of the well). For scale, the red, dashed line shows the mean length-wise *distance* per bout for unflagged bouts. Error bars are s.e.m. (c) The mean speed, change in heading and duration of the bouts (black, red error bars for s.d, s.e.m respectively) from *b* and *O* bout types that are part of the Flagged(ends) sequences from (b). (d) Superimposed images for four random samples of *b* and *O* bout types from the flagged sequences in (a). The green and red are the head positions at the beginning and end of the bout respectively. (e) Superimposed images for four random samples highlighting the Flagged(ends) sequences (blue dots), which include the three bouts before (green dots) and after (red dots) the flagged sequence. Note that the depicted gradient is illustrative.

environment (Fig 6a), though no such selection was explicitly imposed *a priori* in our analysis. To verify that these motifs were indeed the ones involved in aversive chemotaxis, we computed the distance per bout the fish travels in the direction down the aversive gradient for bouts within motifs flagged as over-represented and the rest of the bouts. We find a highly significant bias for the over-represented bouts for swimming down the gradient (Fig 6b). Strikingly, when bouts from the flagged sequences that begin at the two extreme quarters of the well excluding *fTff* are considered, the bias greatly increases, even beyond the typical length-wise distance travelled per bout, indicating that these bouts are triggered by the sharp acidic gradient and are almost exclusively aimed down the gradient. In contrast, the motif *fTff*, which was also reliably observed across fish, was not directed away from the aversive gradients nor did it preferentially occur close to the gradient (see S3 Movie for a sample of the motif *fTff*). These observations suggest that *fTff* may be induced by a global acidification of the swim arena but not as a direct response to the acidic gradient, or is a false positive emerging from stochasticity in the analysis pipeline (e.g., the clustering of bouts). Indeed, we did not observe the *fTff* motif when the entire analysis pipeline including the clustering of bouts was re-run, in support of the latter possibility.

Further examination of the bouts from the *b* and *O* bout types implicated in chemotaxis showed that both bout types were significantly longer and the fish swam faster compared to the unflagged bout types (Fig 6c and S2 Fig). Inspection of the tail movements from this subset of *O* bouts showed that these bouts typically consisted of a large-angle avoidance turn followed by a long burst swim (see Fig 6d for examples). The induced sequences of fast bouts composed of *b* and *O* near the edges of gradient are distinct from the recently described slow avoidance response to $CO_2$ [47]. The previous report of a lack of behavioural response to HCl pH = 4.5 [47] suggests that *b* and *O* may be elicited by more acidic pH.

### Spiraling motifs in soaring trajectories

We note that BASS receives no information about the stimulus experienced by the animal. To confirm that BASS can extract relevant aspects of behaviour even when the cues in the environment are unknown and the stimulus-response map is complex, we turn to simulations of a soaring glider using thermals (updrafts) to gain height in a convective environment [25, 26]. The purpose of this exercise is to present an example of how BASS could be applied to naturalistic behavioural data obtained in the field. We used a dataset of size within experimental reach and input variables that can be readily measured using existing instrumentation.

We simulated trajectories of a soaring glider executing a thermalling strategy in an environment that contains updrafts and downdrafts of varying strengths. The thermalling strategy was learned using reinforcement learning in the field [26]. The glider takes mechanical cues (i.e., torques and accelerations) as input and responds by modulating its bank angle. Past simulations in a turbulent convective environment have shown that a glider executing this thermalling strategy successfully gains height using thermals and exhibits spiralling trajectories similar to those displayed by birds [25]. A sample trajectory is shown in Fig 7a, which shows the glider's meandering path before finding the updraft and the spiralling trajectory while ascending the thermal.

We simulated 75 episodes each (250 seconds per episode) of the glider executing the thermalling strategy and the glider executing a random policy (i.e., it increases, decreases or keeps the same bank angle with equal probability). Units are defined using realistic parameters for glider aerodynamics; for instance, the glider has an airspeed $\sim$10m/s and takes actions every 1.5 seconds (see refs [25, 26] for full details of the simulation). BASS receives as input solely the

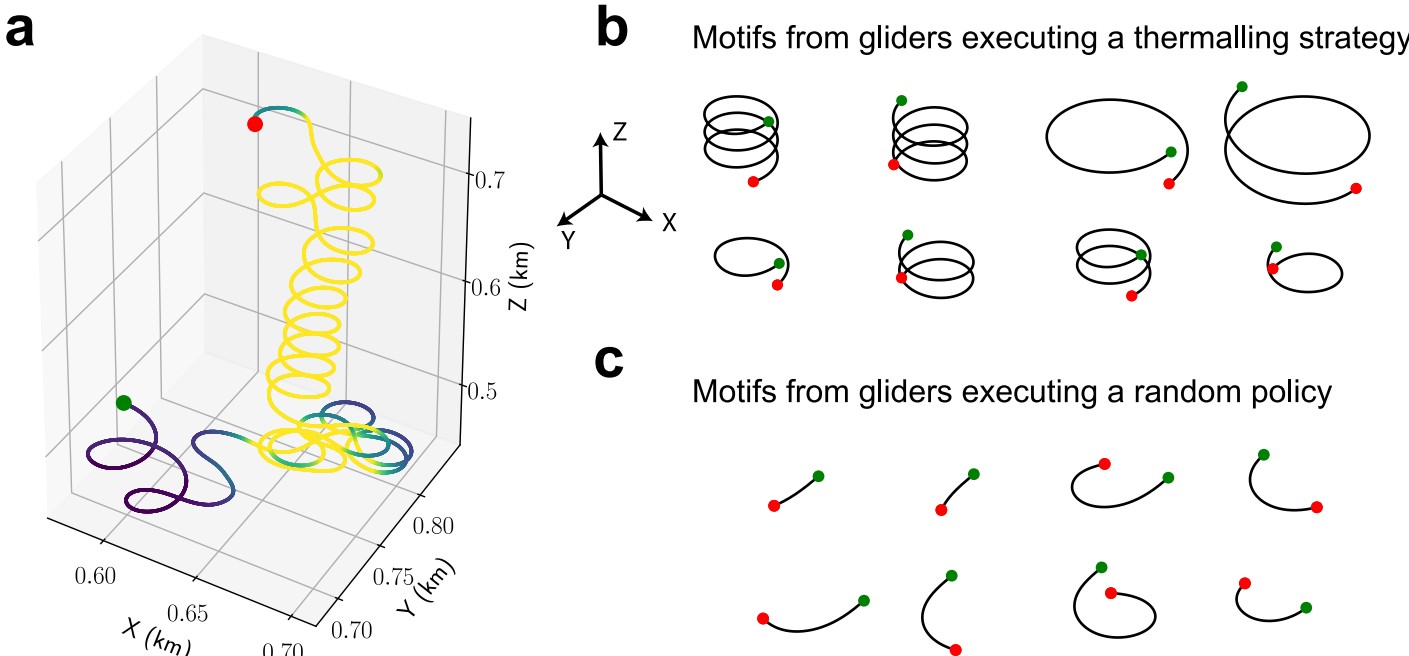

**Fig 7. BASS finds spiralling motifs in simulated trajectories of a soaring glider executing a thermalling strategy.** (a) A sample trajectory showing the thermalling behaviour of the soaring glider. The glider reacts to mechanical cues such as torques and accelerations. See [25, 26] for more details. Regions of downdraft and updraft (blue through yellow) are marked, which highlight the relatively disperse trajectories in downdrafts compared to the spiraling behaviour in updrafts. (b) Motifs found by BASS that span the largest fraction of the thermalling dataset (i.e., episodes where the glider used a thermalling strategy). Most enriched motifs are those of a spiralling pattern. The green and red dots mark the start and end points. Note that the glider sinks without updrafts. (c) Same as (b) for motifs found in the dataset of a glider executing a random policy.

bank angle at every 1.5 seconds, which is divided into five states corresponding to bank angles of $-30°$, $-15°$, $0°$, $15°$, $30°$.

BASS converged to dictionaries with 80 and 65 motifs for the thermalling and random policy datasets respectively ($\epsilon_p = 0.1$, $p_d = 0.5$). In Fig 7b and 7c we show examples of the eight motifs that spanned that largest fraction in the two dataset (i.e., the motifs were sorted according to the number of times they occur in the dataset multiplied by the length of the motif). BASS indeed finds a consistent spiraling pattern in the thermalling dataset, with the longest two motifs of opposite helicity (first two in the top row of Fig 7b) lasting 24 actions, i.e., 36 seconds. Full circles with a lower bank angle and spiraling patterns of shorter length were also found. The motifs from the random policy dataset were roughly half as long (mean length ≈ 8s) on average compared to the motifs from the thermalling dataset (mean length ≈ 16s) and largely consist of short turns as shown in Fig 7c.

## Discussion

In this study, we develop a method to extract recurring action sequences that rarely occur in noisy behavioural data from freely-moving animals. We present a lexical model of animal behaviour, where we view observed behaviour as a composition of recurring motif templates drawn from a dictionary. We develop the BASS algorithm for performing inference on this model, which ultimately yields a dictionary of motifs that the animal performs in its designated environment. Applying the method on data from exploring zebrafish larvae revealed a long time-scale organization of bout sequences that cannot be explained in a Markovian model on single bouts. In a novel aversive chemotaxis task, we identified action sequences of bouts that the fish employ to avoid a noxious environment. When applied to simulated thermalling dataset, BASS succeeded in identifying the long spiraling patterns characteristic to soaring.

BASS is a useful tool to identify behavioural action sequences despite being rare and transient in naturalistic settings when animals are freely moving. To summarize, BASS can be used: 1) when the sensory information perceived by the animal is unknown / uncontrolled, 2) when the behavioural responses is complex and last much longer than a typical locomotor episode, 3) when the behavioural response consist of recurring sequences that occur rarely in the behavioural recording.

We argue that our lexical model yields complementary insight to traditional dynamical models, and is better suited for comparative behavioural analyses, particularly for comparisons across animals in different environments and/or closely related genetic variants. The generative model of independent motifs is the simplest one and fits better to our behavioural data than a Markov model with a similar number of parameters, which is often used to depict quantitative ethograms.

### Relationship with other methods and extensions

Our model generalizes previous work [35–39] on motif discovery from bioinformatics, machine learning and time series analysis by incorporating two important generative processes crucial for behavioural modeling that were not modeled previously: elevating motif templates to latent variables and introducing a data-generating process for elementary locomotor episodes. This generalization is necessary to take into account the significant variability in behavioural data, where clusters in postural space are not always well-defined and erratic movements are common. Note that the generative process for *motif templates* can also be viewed here as a particular hidden Markov model, where the $|\mathcal{D}| + 1$ hidden states at the topmost level are the motifs in the dictionary and the 'background'. In this picture, the full generative model has three levels of hierarchy.

We note that BASS stripped to its most basic form (i.e., with no pattern noise or syllable noise) is simply an efficient method to implement an $n$-gram model for large $n$. One major advantage of BASS is the automatic selection of $n$ based on statistical significance and the capability to find sequences of length beyond what could be feasibly found using a straightforward implementation of an $n$-gram model. Indeed, for the larval zebrafish analysis above, the relatively short abnormal motifs in Table 2 can also be found using an $n$-gram model. However, the analysis using BASS suggests that any other longer, relevant sequences are unlikely to be found in the given dataset. The second advantage of the current formulation lies in the possibility for extending the model along various directions. Prior knowledge can be easily applied. For instance, priors on the distributions of motif lengths or the distribution of frequencies can be introduced by weighting different partitions or a Dirichlet prior on the probabilities, respectively. In both cases, an MLE equation (or a MAP estimate in the latter case) similar to Eq (2) can be derived. To obtain a MAP estimate, the free energy for BASS is modified as $F' = F + C(\{p_m\})$, where $C(\{p_m\})$ is an additional cost term that regularizes the solution or imparts external knowledge. More complex, hierarchical models over motifs may be learned by noticing that the Markovian structure of the partitioning is compatible with structured variational approximations [50]. Further structure can be introduced into the 'background', which we have assumed is made of independently drawn characters, similar to those used in bioinformatics [51]. Finally, unlike $n$-gram models, the BASS framework incorporates pattern noise as a generative model from motif templates to motif realizations. This generative model can be flexibly modified based on the specific pattern noise model most appropriate for the system under consideration without changing the optimization procedure and the basic structure of the model.

Compression methods [52–54] optimize an altogether different "coding" objective, which does not necessarily lead to meaningful motifs; for example, the two-symbol word *ab* could be identified as a motif simply because *a* and *b* occur often, even if *a* and *b* occur next to each other purely by chance. To illustrate this point, we applied the Sequitur (Nevill-Manning) algorithm [52] using PySequitur [55] on the zebrafish larva exploration dataset. Since Sequitur takes a sequence of symbols as input, we collapsed the soft clusters obtained using the GMM to the most likely cluster assignment. Sequitur found a total of $\sim$2500 codewords, much larger than the dictionary obtained by BASS (81 motifs for the hard cluster case). Importantly, the codewords found by Sequitur are not consistent across subsets of the exploratory dataset. Specifically, we split the exploration dataset into two subsets of equal sizes and applied Sequitur to each subset. Only $\sim$23% of the codewords found in the first half were also present in the second half, in contrast to $\sim$61% using BASS (the same parameters as the previous analysis were used except for the hard cluster assignments). This inconsistency is further highlighted when Sequitur is applied to a dataset containing a sequence of 10,000 independently drawn fair coin tosses. BASS found no motifs whereas Sequitur found on an average $\sim$300 codewords despite the random draws. Additionally, the codewords found in two independent realizations of size 10,000 overlap only by 6%, suggesting that codewords found using Sequitur do not represent intrinsic features of the data generation process but are rather sensitive to the specific realization.

As noted above, BASS can be viewed as a constrained version of an hierarchical hidden Markov model (HHMM) with $|\mathcal{D}| + 1$ hidden states. The constraint that each higher-order hidden state corresponds to a motif template, i.e., a *fixed* sequence of symbols rather than a stochastic sequence enables an implementation linear in the dataset size. An HHMM without this constraint has complexity $SL^3$ [56] ($L$ is the size of the dataset and $S$ is the number of states), which makes straightforward implementations infeasible for most practical applications. Linear-time implementations using approximate inference techniques have been

developed [57, 58] and offer promising alternatives. However, the unconstrained HHMM has $O(K^2)$ parameters ($K$ is alphabet size) for each higher-order hidden state, which are to be inferred. Whether the large number of parameters in an HHMM can be inferred from behavioural datasets with many behavioural modes remains an open question and is beyond the scope of the current work.

In certain cases, behaviour is better described as continuous motor output in contrast to the discrete bouts that comprise zebrafish locomotion (e.g the eigenworms that describe *C.elegans* dynamics [43]). In such cases, locally linear approximations similar to those in ref. [18] can be used. The approach there is to fit short windows of dynamics using linear models and then use hierarchical clustering to cluster models in model space into discrete states. A similar approach can be used to generalize BASS to continuous motor output. In particular, we may define the elementary locomotor episodes (via $q(\boldsymbol{y}|c)$) as a set of short continuous snippets (such as lines and curves) equipped with an appropriate probability measure to accommodate imperfect fits. Further, motifs from the extracted dictionaries can be hierarchically clustered to define broader behavioural classes of motifs. These generalizations will be explored in future work.

It is important to emphasize that BASS receives no explicit information about the stimulus experienced by the animal; the extracted motifs therefore do not contain information about the precise stimulus-response map of the animal, but can reveal relevant qualitative aspects of the animal's behaviour when the sensory information is not known. As illustrated here with our simulation, BASS may discover the spiraling of a soaring bird [59] with no reference to what stimulus triggers those responses. More generally, BASS is applicable when characteristic patterns of behaviour are repeated on several occasions during a task. However, it is possible that a single stimulus-response map may yield a highly diverse set of outcomes based on diversity in input and memory of past inputs. In this case, inference is difficult for any algorithm since little statistical information is available without access to the input stream and other methods should be devised.

## Action sequences implicated in aversive chemotaxis

Using BASS, we reveal here that freely-swimming zebrafish larvae navigating in a gradient of aversive cues exhibit recurring sequences of fast bouts mixing burst and large-angle avoidance turns to avoid the aversive environment. Notably, these sequences make up only a very small fraction of the dataset ($\sim 0.2\%$), yet are successfully captured in our analysis.

Furthermore, we discover recurring sequences deployed by zebrafish larvae when exploring arenas of diverse geometry, either rectangular (this study) or square [16]. In all cases, larval zebrafish repeated bout types belonging to the same speed regime (slow or fast). Long repeats of the same bout type, particularly forward and turn swims, occur frequently for $\sim 3$–$8$ iterations. This observation is consistent with prior observations of repetition of turns in freely-swimming larvae [46]. In the baseline conditions when the environment contains no aversive cue, larvae mainly display motifs consisting of slow forward bouts and turns. In contrast, larval zebrafish exposed to noxious stimuli, such as the acidic gradients that activate cutaneous sensory afferents, respond using transient action sequences of fast forward and/or turns in order to swim away. While an automated categorization of bouts can identify that large-angle avoidance turns $O$ often occur in the acidic gradients (S3 Fig), our method reveals a richer chemotactic response composed of fast locomotor episodes including burst swims and large angle turns. Remarkably, BASS led to the discovery of a very interesting phenomenon that is preserved across datasets from multiple labs operating with different zebrafish strains and different arenas. We show that larval zebrafish perform highly-specific sequences containing either only slow (during basic exploration) or only fast (during aversive chemotaxis) locomotor

episodes. This observation suggests that the descending command signals sent to the spinal cord to elicit forward bouts [60] or turns [61] last over seconds to tens of seconds, possibly via sustained inputs to reticulospinal neurons in the hindbrain [60]. The dichotomy in terms of speed regime observed within sequences of locomotor episodes may therefore have important implications for the construction of neuronal networks sustaining foraging and avoidance.

## Conclusion

BASS can be easily incorporated into existing behavioural analyses pipelines alongside the expanding repertoire of methods established for unsupervised behavioural clustering that deal with classifying individual locomotor episodes [1–5]. In its current form, our implementation can handle datasets of size ≲ 300,000 bouts and dictionaries of size ≲ 500, beyond which approximations for scalable inference have to be developed. Our approach further highlights the connection between behavioural modeling and genomics, where a wealth of algorithms have been developed. Exploiting this analogy may lead to a fruitful exchange of techniques between the two seemingly-disparate fields. Finally, we remark that our method is an addition to a rapidly enlarging computational toolkit for extracting mechanistic answers to behavioural questions. BASS complements existing computational methods for unsupervised behavioural clustering, typically used to segment individual locomotor episodes, by finding structure in the form of recurring sequences of locomotor episodes.

## Materials and methods

### Ethics statement

Animal handling and behavioural assays on 6–7 days old larvae were carried out with the validation of the Institut du Cerveau (ICM) in agreement with the French Ministry for Research and Education and the French National Ethics Committee (Comité National de Réflexion Ethique sur l'Expérimentation Animale, APAFIS #2018071217081175) following the European Communities Council Directive (2010/63/EU). All zebrafish animals were reared and maintained at +28.5˚C on a 14/10 hour light/dark cycle until 7 days post fertilization (dpf) when behavioural experiments were conducted.

### Behavioural assays and recording

Wild-type AB zebrafish were reared at the density of 20 larvae per petri dish filled with 30 mL of E3 medium. At 4 dpf, the E3 medium was replaced to avoid waste accumulation. On the day of behavioural recording, ten minutes prior to the beginning of experiments, all larvae were transferred onto the LED illumination plate to allow habituation to the lighting of the setup. At the beginning of each trial, we positioned on the illumination plate the recording chamber containing 12 swim arenas (length 140 mm x width 10 mm x height 4 mm) filled with 4 mL of E3 medium. Each individual larva was then transferred into the central region of a swim arena delimited by a custom-made comb. Once all larvae were positioned, the comb was carefully removed to enable animals to explore freely the arena and the video recording was launched.

To monitor the navigation in gradients of pH, larvae were exposed to acidic solution at both ends of the swim arenas. To create this double acidic gradient, an acidic solution was prepared by diluting 1 M HCl (Merck, #109057) 1:6 into E3 medium (final pH 1.5). 100 $\mu$L of this solution was applied at both ends of the swim arena before removing the comb and launching the recording. Out of 12 swim arenas, 6 were filled with the double gradient of acidic pH and 6 were filled with E3 medium only. Before any behavioural recordings, we estimated the stability of the pH gradient generated by following the same application protocol and monitoring the

diffusion of red food colorant over 10 minutes. These pilot experiments indicated that in our conditions, the acidic solution slowly diffused and formed a steep gradient about 20 mm from each end of the swim arena over the 10 min-long recording.

Ten min-long videos were recorded at 160 Hz with an exposure time of 1 ms, and a pixel size of 70 $\mu$m using a ViewWorks camera (Basler acA2040–180km) controlled by the Hiris software (R&D Vision, Nogent sur Marne, http://www.rd-vision.com/r-d-vision-eng).

## Tracking

To track the head and tail positions of zebrafish larvae, we used the open-source software ZebraZoom (https://zebrazoom.org/) [41, 62]. The algorithm begins by locating all the wells and by extracting the background of the video. ZebraZoom first applies a series of actions to detect the animal in each well: i) contours of head and entire body are detected using active contours, ii) the center of the head is identified as the center of mass of the head contour and the tip of the tail is detected using both the curvature along the body contour and distance to the center of the head. The midline is then identified between the left and right borders of the body contour. For each animal, the difference in pixel intensity between subsequent frames enables the automated detection of bout start and end. Then, for each bout, the algorithm calculates the head position, head direction and the tail angle from which kinematic parameters are subsequently estimated: number of oscillations, instantaneous tail beat frequency, maximum amplitude for each tail bend, bout speed, bout duration, and bout distance. Tunable parameters in the tracking algorithm were optimized to detect small amplitude forward bouts occurring frequently during exploration. In order to validate our algorithm, we manually inspected validation videos where the head direction and tail position were superimposed on the raw image when a bout is detected, allowing to check both the tracking and bout detection quality.

## Bout categorization

A segmented bout contains the positions of the head and ten tail segments for each frame. For each bout, we computed the average speed, the change in heading angle and the duration of the bout. The tail angle relative to head orientation of the seven tail segments farthest away from the head were computed and the first 16 frames of these 7 quantities were concatenated to form a 112-dimensional vector. Principal Component Analysis (PCA) on this vector yielded a projection onto 3 principal components that together explained 80% of the variance. For bout categorization, the average speed, the change in heading and the summed tail angle (integrated absolute value of tail amplitude) were combined with the first 3 tail angle principal components, which together served as the six-dimensional input (denoted $y$ throughout the paper) for the clustering algorithm. Including other kinematic variables such as tail maximum amplitude, inter-bout interval, duration, number of oscillations, angular speed or displacement did not change our bout categorization and were excluded for the rest of the analysis.

For clustering of bouts, we employed a Gaussian Mixture Model (GMM) with a full covariance matrix. We modified the standard expectation-maximization algorithm to fit multiple datasets simultaneously such that the emission functions i.e., the probability of the input given cluster label, $q(y|c)$, was fixed for all datasets but the cluster weights differ across datasets. To do this, we first sub-sampled the datasets so that they all have the same size. The objective function to be maximized for the GMM model was modified to be the sum of the log-likelihoods for each dataset. We found that adding a term proportional to the Jensen-Shannon divergence between the cluster weights across the two datasets encouraged the algorithm to find cluster assignments whose weights differed the most across datasets. The number of clusters was set

to seven by measuring the log-likelihood on a held-out dataset. The distributions of the speed, duration and change in heading for different bout types are shown in S2 Fig.

We performed a similar procedure to analyze the data from Marques et al [16]. The dataset contained approximately 120,000 bouts from 23 zebrafish larvae at 6–7 dpf imaged individually at 700 frames per second at a resolution of 62 microns per pixel with different light intensities. From the raw head position data, for each bout we extracted the displacement and change in heading in the first 120 frames (170 ms). Tail angles for five tail segments (3rd furthest away from the head to the 2nd closest) sub-sampled at half the frame rate were concatenated together to form a 300-dimensional vector. PCA yielded four principal components, which were combined with the displacement and change in heading to form the input to the GMM (S4 Fig). More clusters simply yielded finer divisions of the clusters. In their paper, Marques et al use a density-based clustering algorithm to identify 13 different bout types. In this dataset, most bouts corresponded to AS, Slow 1, Slow 2, Routine Turns with a small but significant fraction of SATs and HATs (which together formed the first four bout types *f*, *F*, *t* and *T*) and a much smaller fraction of other bout bout types. Most of the rare bout types were not identified in our clustering and were part of the other (*O*) bout type.

## Modeling

In this Section, we expand on the model presented in the main text. We first develop the model followed by the method to build the dictionary.

**Generative model.**  Given a time series of length $L$, $\boldsymbol{Y} = \boldsymbol{y}_1 \, \boldsymbol{y}_2 \ldots \boldsymbol{y}_L$, a clustering algorithm with $K$ clusters (labeled from 1 to $K$) applied on each $\boldsymbol{y}_i$ converts the dataset $\boldsymbol{Y}$ into a sequence of probability vectors, $\boldsymbol{R} = \boldsymbol{\rho}_1 \, \boldsymbol{\rho}_2 \ldots \boldsymbol{\rho}_L$, where each $\boldsymbol{\rho}_i$ is a $K$-dimensional probability vector. To fix notation, the $j$th component of this vector, $\boldsymbol{\rho}_{ij}$, is given by $\boldsymbol{\rho}_{ij} \equiv q(\boldsymbol{y}_i|j)$, where $q(.|j)$ is the probability density function associated to cluster label $j$. We use cluster label and characters of an alphabet interchangeably. Each character represents a well-defined locomotor episode; in the case of larval zebrafish, this is the category of a single bout. In the case of the soaring glider, the glider takes discrete actions of changing the bank angle every 1.5 seconds. For each action, the glider changes its bank angle linearly over 1.5 seconds for a total change of 0˚ or ±15˚ starting from one of five angles: −30˚, −15˚, 0˚, 15˚, 30˚. The bank angle at the beginning of the action is taken as the postural state. The appropriate choice of the timescale used to define a postural state is organism-dependent. We assume the timescale and the clustering scheme have been chosen appropriately based on prior knowledge about the organism. We do not address the details of how to find an appropriate clustering scheme for each scenario; we refer to reviews on the topic [2, 4, 5].

We define a motif $\boldsymbol{m} = c_1 \, c_2 \ldots c_l$ as a string of characters (or cluster labels) of arbitrary length $l$ ($l$ is not a parameter), which belongs to a dictionary $\mathcal{D}$ that contains all the motifs. Each motif in the dictionary and each character in the alphabet are assigned probabilities $p_{\boldsymbol{m}}$ which are to be inferred from data. Henceforth, we will use 'templates' to mean both motifs and characters. We assume the data is generated as follows: First, templates are generated sequentially, drawn independently and identically from the distribution $p_{\boldsymbol{m}}$. Second, in each instantiation, the template has a probability of 'mutating' to a closely-related pattern. The model for mutation of templates is described in the following paragraph. Third, data is generated from the mutated templates character-by-character from $q(\boldsymbol{y}|c)$. For instance, if the mutated template is given by $\tilde{\boldsymbol{m}} = \tilde{c}_1 \tilde{c}_2 \ldots \tilde{c}_{\tilde{l}}$, a data vector of length $\tilde{l}$, $\boldsymbol{y}_1 \boldsymbol{y}_2 \ldots \boldsymbol{y}_{\tilde{l}}$, corresponding to $\tilde{\boldsymbol{m}}$ is drawn from the distribution $\prod_i q(\boldsymbol{y}_i|\tilde{c}_i)$.

Now, we define the likelihood function for motifs $Q(\boldsymbol{Y}|\boldsymbol{m})$, where $\boldsymbol{Y}$ is a sequence of data vectors $\boldsymbol{y}_1 \boldsymbol{y}_2 \ldots \boldsymbol{y}_{\tilde{l}}$ and $\boldsymbol{m} = c_1 \, c_2 \ldots c_l$. A character in a motif can be mutated in two ways: a

deletion, where the character does not appear, or copy insertions, where the character is copied in-place a certain number of times. This is similar to the operations allowed in a profile HMM. We do not allow for transpositions of two characters or insertions of new characters that are not identical to the one already present. To give an example, a motif *abb* ($K = 2$ alphabet size) may mutate to *ab* or *bb* corresponding to one deletion, and *aabb* or *abbb* corresponding to one insertion, but cannot mutate to *abab* via one insertion or deletion (see main text for an explanation for this choice). We assume each character $m_i$ in the motif has a probability $\epsilon_p p_d$ of being deleted, probability $\epsilon_p p_{c,j}$ of being copied exactly $j$ ($j > 0$) times and is neither deleted nor copied with probability $1 - \epsilon_p$. To normalize, we require $p_d + \Sigma_j p_{c,j} = 1$. We can write down a recursive formula for computing $Q(\boldsymbol{Y}|\boldsymbol{m})$. Intuitively, we start from the last character in the motif and assign it a certain number of data points depending on whether there is a deletion (0 points), $j$ insertions ($j + 1$ points) or neither (1 point). After the assignment, the last character and all the data points assigned to it are removed, and the process is repeated until there are no data points left to be assigned. Denote $Q(\boldsymbol{y}_1 \boldsymbol{y}_2 \dots \boldsymbol{y}_{k_1} | c_1 c_2 \dots c_{k_2}) \equiv M(k_1, k_2)$, where $0 \le k_1 \le L$ and $0 \le k_2 \le l$. For $k_1, k_2 > 0$, $M$ can be computed recursively as

$$M(k_1, k_2) = (1 - \epsilon_p) q(\boldsymbol{y}_{k_1} | c_{k_2}) M(k_1 - 1, k_2 - 1) + \epsilon_p p_d M(k_1, k_2 - 1)$$

$$+ \sum_{j=1}^{k_1} \epsilon_p p_{c,j} M(k_1 - j, k_2 - 1) \prod_{k'=0}^{j} q(\boldsymbol{y}_{k_1 - k'} | c_{k_2}), \tag{3}$$

with boundary conditions $M(0, k_1) = 0$ for $k_2 > 0$ and $M(0, 0) = 1$. In the applications presented in this paper, we set $p_{c,1} = 1 - p_d$ non-zero and the rest zero (i.e., only one copy of a character is allowed).

**Likelihood function and proof that normalization is unity.** Given a dataset $\boldsymbol{Y} = \boldsymbol{y}_1 \boldsymbol{y}_2 \dots \boldsymbol{y}_L$, its likelihood under the model is given by the sum over all possible $2^{L-1}$ partitionings $\{\pi\}$ of the dataset, where *partition $\alpha$* in *partitioning $\pi$* is denoted $\boldsymbol{Y}_\alpha^\pi$. Each partitioning is weighted by its likelihood. We have

$$P(\boldsymbol{Y}; \{p_{\boldsymbol{m}}\}) = Z'^{-1} \sum_\pi \prod_{\alpha=1}^{N(\pi)} Q(\boldsymbol{Y}_\alpha^\pi), \tag{4}$$

where the marginal probability is $Q(\boldsymbol{Y}_\alpha) = \Sigma_{\boldsymbol{m}} Q(\boldsymbol{Y}_\alpha | \boldsymbol{m}) p_{\boldsymbol{m}}$, $N(\pi)$ is the total number of templates in partition $\pi$ and $Z'$ is the normalization constant, which is integrated over all possible $\boldsymbol{Y}$. The product of the marginals $Q$ in the above equation is the likelihood of each partitioning. We give a simple (non-rigorous) argument for $Z'$ being close to unity for large $L$. Intuitively, this is because the normalization should be one if not for boundary effects, which we show are negligible for large datasets. We have

$$Z'(L) = \int \mathcal{D}\boldsymbol{Y}_L P(\boldsymbol{Y}_L; \{p_{\boldsymbol{m}}\}) \tag{5}$$

where the subscript is introduced to keep track of its length. For any $l$, conditioning on the first partition before $l$, we have the recursive formula

$$
\begin{aligned}
Z'(l) &= \sum_{i=1}^{l-1} \int \mathcal{D}\boldsymbol{Y}_i Q(\boldsymbol{Y}_i) Z'(l - i) \\
&= \sum_{\boldsymbol{m}} p_{\boldsymbol{m}} \sum_{i=1}^{l-1} \int \mathcal{D}\boldsymbol{Y}_i Q(\boldsymbol{Y}_i | \boldsymbol{m}) Z'(l - i)
\end{aligned}
\tag{6}
$$

Consider the case when $l \ll L$ and we modify the sequence so that the first $l - 1$ points in the sequence are replaced by 1s instead of the appropriate integrals over the marginals $Q$ i.e., $Z'(i) = 1$ for $i < l$. Since $l \ll L$, we must have $Z'$ for this modified sequence approximately equal to the original $Z'$, with corrections being $O(l/L)$. By the above recursive formula, for this modified sequence $Z'(l) = 1$ because of the normalization of $p_m$ and $Q(Y|m)$ and it follows that $Z'(l') = 1$ for any $l' > l$. An approach for a rigorous argument could be to find an upper bound for $|1 - Z'(i)|$ for $i < l \ll L$ and proceed similarly.

**Maximum likelihood estimation of motif probabilities.** Since $Z' \simeq 1$, $\{p_m\}$ enter the equation only via $Q(Y_\alpha)$. We use a Lagrange multiplier $\lambda$ to impose the normalization constraint $\sum p_m = 1$ and maximize $P(Y;\{p_m\})$. Taking the derivative w.r.t $p_m$ and equating to zero, we get

$$\lambda = \sum_\pi \sum_{\alpha'} Q(Y_{\alpha'}^\pi | \boldsymbol{m}) \prod_{\alpha \neq \alpha'} Q(Y_\alpha^\pi). \tag{7}$$

Multiplying on both sides by $p_m$ and using Bayes' rule, $Q(Y_{\alpha'}^\pi | \boldsymbol{m}) p_m = Q(Y_{\alpha'}^\pi) p(\boldsymbol{m} | Y_{\alpha'}^\pi)$, we get the implicit equation for the MLE,

$$\lambda p_m^* = \sum_\pi \sum_{\alpha'=1}^{N(\pi)} p(\boldsymbol{m} | Y_{\alpha'}^\pi) \prod_{\alpha=1}^{N(\pi)} Q(Y_\alpha^\pi). \tag{8}$$

The sum over the posterior probabilities can be interpreted as an effective number of counts of $\boldsymbol{m}$ in the partition $\pi$. We then have $p_m^* = \langle N_m \rangle / \sum_{m'} \langle N_{m'} \rangle$, where $\langle N_m \rangle$ is the expected number of counts of $\boldsymbol{m}$ over the ensemble of partitions.

**Numerical optimization.** To compute $p_m$ numerically, we define the free energy, $F = -\ln P(Y;\{p_m\})$ and minimize it under the constraint $\sum_m p_m = 1$. We impose this constraint using the transformation $p_m = \frac{e^{-\beta_m}}{\sum_{m'} e^{-\beta_{m'}}}$, fixing one of the $\beta_m$'s to zero and optimizing for the rest. We use the L-BFGS-B algorithm, which requires $F$ and the gradient of $F$, whose calculation is described below. The Hessian can be computed exactly and Hessian-based methods applied, but we found that the gradient-based methods work equally well and are in fact slightly faster. Note that since $\langle N_m \rangle = -p_m \partial_m F$, the gradient calculation can also be used to calculate the number of occurrences of a motif.

**Free energy and gradient calculation.** The calculation of $F$ and $\partial_m F$ is performed using transfer matrix-like techniques. The procedure is similar to the one described in ref [35] (see [63] for technical details). The key insight from ref [35] is that the partitioning of the data vector has a Markovian structure, which allows calculations similar to the forward-backward algorithm for Hidden Markov Models. The same structure is preserved in our model albeit with the added complication that the motifs themselves are latent variables and thus all possibilities need to be considered for each partition, which is taken into account in the marginal $Q(Y_\alpha)$ in (4). The procedure described below holds for a single unbroken sequence. In practice, the dataset contains a collection of sequences from different sources and the procedure is easily adapted to the multiple sequence case. Below, we use a subscript $i:j$ to mean the quantity is evaluated for the sequence $y_i y_{i+1} \ldots y_j$. Let $Z_{i:j}$ denote $P(Y_{i:j}|\{p_m\})$. We can write the recursive formula

$$Z_{1:i} = \sum_{l=1}^m Q_{i-l+1:i} Z_{1:i-l}, \tag{9}$$

where the marginal probability in shorthand is $Q_{i-l+1:i} = \sum_m Q_{i-l+1:i}^m p_m$ and $Q^m$ is the motif likelihood function of $\boldsymbol{m}$, $Q(.|\boldsymbol{m})$. Here we truncate the length of the partition to $m = 2l_{\max}$,

where $l_{\max}$ is the length of the longest motif since we allow at most one duplication per character.

In practice, the $Q$ values become extremely small and there is underflow. We instead calculate the ratio $R_i = Z_{1:i}/Z_{1:i-1}$, for which we can write

$$R_i = Q_{i:i} + \sum_{l=2}^{m} Q_{i-l+1:i} \left( \prod_{k=i-l+1}^{i-1} R_k \right)^{-1}. \qquad (10)$$

The free energy is $F = -\sum_{i=1}^{L} \ln R_i$. For the derivative, it is also useful to calculate the quantity $R'_i = Z_{i:L}/Z_{i+1:L}$, which is calculated as

$$R'_i = Q_{i:i} + \sum_{l=2}^{m} Q_{i+l-1:i} \left( \prod_{k=i+1}^{i+l-1} R'_k \right)^{-1} \qquad (11)$$

The derivative $\partial_m F$ is calculated using

$$\partial_m F = -\sum_{i=1}^{L} G_{i-l+1,i} Q^m_{i-l+1:i}, \qquad (12)$$

where $G_{i-l+1,i} \equiv Z_{1:i-l} Z_{i+1:L}/Z_{1:L}$ is calculated using

$$G_{i-l+1,i} = \left( \prod_{k=1}^{i-l} R_k/R'_k \right) \left( \prod_{k=i-l+1}^{i} R'_k \right)^{-1}. \qquad (13)$$

In practice, calculating $Q^m_{i-l+1:i}$ is the bottleneck, and we pre-calculate and store non-zero (above a threshold) values of $Q^m_{i-l+1:i}$ before the optimization for $p_m$ is done. Finally, gradients in $p_m$ are converted to gradients in $\beta_m$ for optimization.

**Dictionary expansion.** As described in the main text, new motifs are added by considering all possible concatenations of pairs of motifs in the dictionary. Consider the motif $mm'$ obtained by two existing motifs $m$ and $m'$. The expected number of counts in $Y$ of $mm'$ is calculated using the derivative of $F$:

$$\langle N_{mm'} \rangle = \zeta(mm') \sum_{i=1}^{L} G_{i-l+1,i} Q^{mm'}_{i-l+1:i}, \qquad (14)$$

where $\zeta(mm')$ is the probability under the model of all possible partitionings of the concatenated motif $mm'$. For example, if the dictionary contains $a$, $b$ and $ab$ with probabilities $p_a$, $p_b$, $p_{ab}$, we have $\zeta(abab) = p_a p_b p_a p_b + p_a p_b p_{ab} + p_{ab} p_a p_b + p_{ab} p_{ab} = p_a^2 p_b^2 + 2 p_a p_b p_{ab} + p_{ab}^2$. $\zeta$ can be computed efficiently using an equation similar to (9). $\langle N_{mm'} \rangle$ is compared to the model prediction $\bar{N}\zeta(mm') \simeq L\zeta(mm')/\bar{l}$, where $\bar{l}$ is the mean motif length, $\bar{l} \equiv \sum_m l_m p_m$. A likelihood ratio test yields a $p$-value and pairs below a threshold (we use 0.001) are added to the dictionary.

**Dictionary truncation.** For truncation, the set $\{p_m\}$ is re-calculated. To prune clusters of similar motifs, we define a distance measure between a pair of motifs using the Jensen-Shannon (JS) divergence of the data distributions $Q$ generated by the pair, calculated using Monte Carlo sampling i.e., the distance between two motifs is

$$d(m, m') = \frac{D_{\mathrm{KL}}(Q(.|m)||Q') + D_{\mathrm{KL}}(Q(.|m')||Q')}{2\ln 2}, \qquad (15)$$

where $Q' = (Q(.|m) + Q(.|m'))/2$ and $D_{\mathrm{KL}}$ is the standard Kullback-Leibler divergence between

two distributions. The factor 1/2ln2 restricts the range of $d$ between 0 and 1. Motifs which have a distance less than a threshold (we use $J_{thr} = 0.15$, see S5 Fig) are considered similar. We re-weight motif probabilities by adding the probabilities of similar motifs to the estimated one, keep the representative motif with the highest re-weighted probability in a cluster and discard the rest. Finally, $\{p_m\}$ is again calculated and motifs which have low counts (we use 5) are discarded. Individual characters are not discarded from the dictionary using the above criteria until the last iteration. Note that pruning using the JS divergence is only performed when $\epsilon_p > 0$; the JS distance is always 1 otherwise.

We keep track of the free energy per symbol of the dataset after every iteration of the above three steps. We find that the free energy monotonically decreases in almost all cases and converges to a local optimum, though we have no formal proof as to why it is monotonic. The iterative procedure is stopped when the relative change in free energy is <0.1% for two consecutive iterations. The procedure usually converges within 15 iterations.

**Most likely sequence of motifs.** The HMM structure of the generative model allows us to compute the most likely sequence of motifs that generate the dataset in a manner similar to the Viterbi algorithm. In particular, given the dataset $Y$, we would like to find the most likely sequence $m_1^*, m_2^*, \ldots$ defined as

$$m_1^*, m_2^*, \ldots = \arg\max_{m_1, m_2, \ldots} P(m_1, m_2, \ldots | Y) \tag{16}$$

$$= \arg\max_{m_1, m_2, \ldots} P(Y|m_1, m_2, \ldots) \prod_i p_{m_i} \tag{17}$$

The max over the quantity on the right hand side can be computed using dynamic programming; we define a value function for the probability of the most likely motif sequence $\max_{m_1, m_2, \ldots} P(m_1, m_2, \ldots | Y_{1:k})$ for the length $k$ sequence $Y_{1:k}$ and iterate over the next most likely motif.

**Generation of the synthetic dataset.** The data is drawn from a Gaussian mixture model with 7 components in two dimensions as shown in Fig 2a of the main text. We sample a frequency distribution over the 7 characters in the alphabet from a symmetric Dirichlet distribution ($\alpha = 5$). We then construct a dictionary of 50 motifs, with each motif having a length drawn from a Poisson distribution with mean 5. The characters in each motif are drawn according to the frequency distribution over the alphabet. The probability of each motif, $p_m$, is drawn from a symmetric Dirichlet distribution ($\alpha = 1$) and scaled with a parameter $1 - \epsilon_b$, where $\epsilon_b$ is the fraction of the dataset that is made up of individual characters and specifies the 'background noise'.

The data is sampled according to the generative process described previously. We use $\epsilon_p$ as a measure of action pattern noise, which is the probability of a deletion or duplication (which are set equal) per symbol in the motif as described previously. To quantify locomotor episode noise, we define the discriminability $\mu$ between clusters using the distance between neighboring clusters (each of which has standard deviation one) as shown in Fig 2a. For the numerical experiments, we set $\epsilon_b = 0.5$ fixed and vary $\epsilon_p$ and $\mu$. In the example shown in main text, we set dataset size $L = 40000$, $\epsilon_p = 0$, $\mu = 3$.

## Supporting information

**S1 Fig. BASS performance on synthetic data with pattern noise.** (a) The true probabilities of the motifs (yellow dots) and probabilities estimated (green dots) by our algorithm showing successful reconstruction of the dictionary for $\epsilon_p = 0.1$, $L = 40,000$. The crosses are low-probability motifs not identified by the algorithm. (b) The percentage of correctly placed partitions with increasing dataset size. Note that this measure differs from the measure used in Fig 2c

(which additionally measures whether the correct symbol is decoded). This new measure is simpler to calculate when $\epsilon_p \neq 0$ since an extra alignment step is required to use the former measure when there are insertions and deletions. The corresponding optimal partitioning performance when the true dictionary is known is shown in dashed lines. In both panels, we use $\epsilon_b = 0.5$, $\mu = 3$, $p_d = 0.5$.
(EPS)

**S2 Fig. Bout categorization.** (a) The held-out log likelihood from fitting the Gaussian Mixture Model to the exploratory and aversive datasets vs the number of mixture components. We characterize each bout using its speed, change in heading, summed tail angle (integrated absolute tail angle over time) and the first three PC components of the tail angles. (b) The histograms of speed, change in heading and duration of bout for each bout type.
(EPS)

**S3 Fig. Bout locations along the well.** Histogram of larvae positions along the well in exploratory (black) and aversive (red) datasets.
(EPS)

**S4 Fig. Bout categorization and motif segmentation of the dataset from Marques et al.** (a) The five bout types identified using a Gaussian Mixture Model. Each bout is characterized using its displacement, change in heading and the first four PC components of the tail angles. (b) The histograms of displacement and change in heading for the five bout types. (c) A sample from the dataset showing the most likely segmentation of the data into motifs from the learned dictionary.
(EPS)

**S5 Fig. Hyperparameter selection via held-out test sets.** Pattern noise $\epsilon_p$, the probability of a deletion $p_d$ and the Jensen-Shannon divergence threshold for removing similar bouts, $J_{\text{thr}}$, were set by computing the free energy per bout on a randomly split held-out dataset of approximately 18000 bouts (training dataset size was 70000). The free energy per bout for each tested parameter triplet is computed by averaging over 18 trials (with a separate random split for each trial). The red square shows the parameters ($\epsilon_p = 0.1$, $p_d = 0.2$, $J_{\text{thr}} = 0.15$) chosen for the rest of our analysis.
(EPS)

**S1 Table. Motifs discovered in exploratory data.** The 25 motifs that deviate most from Markovianity are shown (as measured by a $p$-value with Markovianity as the null hypothesis, see main text).
(PDF)

**S2 Table. Motifs discovered in the data from Marques et al.** The 25 motifs that deviate most from Markovianity are shown (as measured by a $p$-value with Markovianity as the null hypothesis, see main text).
(PDF)

**S3 Table. Motifs discovered in the data from chemotactic fish in the aversive environment.** The 25 motifs that deviate most from Markovianity are shown (as measured by a $p$-value with Markovianity as the null hypothesis, see main text).
(PDF)

**S1 Movie. Movie showing the raw video captured by the camera and the most likely bout categorization using the Gaussian Mixture Model.** The data is from a single larva in the exploratory environment.
(MOV)

**S2 Movie. Movie showing the raw video captured by the camera and the most likely bout categorization using the Gaussian Mixture Model.** The data is from a single larva in the aversive environment.
(MOV)

**S3 Movie. Movie showing a sample of the the abnormal motif *fTff* (see main text).**
(MOV)

## Acknowledgments

We thank Monica Dicu and Antoine Arneau for fish care in the Phenoparc animal core facility of ICM, Maxime Kermarquer for use of the ICM cluster, Kristen D'Elia, Sophia Horowitz and Clara Besserer for trouble-shooting at the early stages of the chemotaxis experiments, Bing Brunton, Elena Rivas and Massimo Vergassola for useful comments, and Joao Marques and Michael Orger for sharing their zebrafish larvae dataset.

## Author Contributions

**Conceptualization:** Gautam Reddy, Laura Desban, Claire Wyart.

**Data curation:** Gautam Reddy.

**Formal analysis:** Gautam Reddy, Hidenori Tanaka.

**Funding acquisition:** Claire Wyart.

**Investigation:** Gautam Reddy, Laura Desban, Hidenori Tanaka, Julian Roussel, Claire Wyart.

**Methodology:** Gautam Reddy, Laura Desban, Julian Roussel, Claire Wyart.

**Project administration:** Claire Wyart.

**Resources:** Gautam Reddy.

**Software:** Gautam Reddy, Olivier Mirat.

**Supervision:** Claire Wyart.

**Validation:** Olivier Mirat.

**Visualization:** Gautam Reddy.

**Writing – original draft:** Gautam Reddy, Claire Wyart.

**Writing – review & editing:** Gautam Reddy, Claire Wyart.

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
