## [Decision Letter · Decision Letter 0]

6 Apr 2021

Dear Dr. Reddy,

Thank you very much for submitting your manuscript "A lexical approach for identifying behavioral action sequences" for consideration at PLOS Computational Biology.

As with all papers reviewed by the journal, your manuscript was reviewed by members of the editorial board and by several independent reviewers. In light of the reviews (below this email), we would like to invite the resubmission of a significantly-revised version that takes into account the reviewers' comments. We would like to draw your attention especially to 1. an appropriate review in the introduction and discussion of your work in relation to the extensive literature in the domain of syntactical understanding of behaviour - which may require toning down or better explaining your innovation and 2. to consider whether your models, given the results of previous publications (see 1.), are powerful enough to capture all results and to evaluate them according to the state-of-the-art. We appreciate that the field is very cross-disciplinary and hence you may have not seen some of these works, however for publication in a cross-biological journal such as PLOS CB, we would expect you to cover these.

We cannot make any decision about publication until we have seen the revised manuscript and your response to the reviewers' comments. Your revised manuscript is also likely to be sent to reviewers for further evaluation.

Sincerely,

Aldo A Faisal

Associate Editor

PLOS Computational Biology

Stefano Allesina

Deputy Editor

PLOS Computational Biology

Reviewer's Responses to Questions

**Comments to the Authors:**

Reviewer #1: "A lexical approach for identifying behavioral action sequences" by Reddy et al. draws on an analogy between behaviour and language to develop a new model of behaviour sequences. The model is related to models first developed in bioinformatics but has been extended to deal with differences between language and DNA sequence data (which are intrinsically discrete) and behavioural data where discretisation is noisy. The authors show they can fit their model with an EM algorithm on simulated data, zebrafish behaviour data, and on simulated glider motion. They identify rare but conserved sequences with interesting properties. For example, they find that slow and fast bouts occur in separate motifs (as opposed to being interspersed with each other). The paper is engaging and clearly written.

I have only two main (interrelated) concerns. The first is that the model is motivated using a contrast with recent methods for behaviour classification, arguing that these methods don't capture behavioural sequences. As currently presented the introduction elides the long history in ethology of trying to capture and analyse behavioural sequences. To give just a few examples that explicitly mention the connection to language:

-The problem of serial order in behavior

KS Lashley - 1951

-Grammar of a movement sequence in inbred mice

JC Fentress, FP Stilwell

Nature, 1973

-Natural syntax rules control action sequence of rats

KC Berridge, JC Fentress, H Parr

Behavioural brain research, 1987

-Linguistic analogies and behavior: The finite-state behavioral grammar of food-hoarding in hamsters

CH Jones, JPJ Pinel - Behavioural brain research, 1990

-A syntax of hoverfly flight prototypes

BRH Geurten, R Kern, E Braun, M Egelhaaf

Journal of Experimental Biology 2010

-Changes in Postural Syntax Characterize Sensory Modulation and Natural Variation of C. elegans Locomotion

RF Schwarz, R Branicky, LJ Grundy, WR Schafer, AEX Brown

PLOS Comp Biol 2015

-Grammars of action in human behavior and evolution

D Stout, T Chaminade, A Thomik, J Apel, A Faisal

bioRxiv 2018

-Drosophila melanogaster grooming possesses syntax with distinct rules at different temporal scales

JM Mueller, P Ravbar , JH Simpson, JM Carlson

PLOS Comp Biol 2019

I realise there is a massive literature here and I'm not suggesting the authors need to write a history of ethology in their introduction, nor am I suggesting this takes away from their new model which is an exciting contribution. Instead, I think it would be useful to frame the introduction slightly differently around the idea that ethology has long been concerned both with identifying behaviours and with understanding their sequences. Much of the recent work in computational ethology (but not all) has focussed on the first problem. The contribution of the current paper is a new model to solve the second problem. But rather than being the first approach to the second problem, it builds on exiting approaches. This should be reflected especially in the second paragraph of the introduction which currently has no citations at all.

My second concern is about actually comparing the current model to some of these earlier sequence analysis methods. The authors use a Markov model as a null model to find over-represented sequences, but given previous work suggesting hierarchical structure in behaviour sequences (e.g. Berman et al. cited in the manuscript), a hierarchical Markov model could be a useful additional comparison.

The authors mention n-gram counting as another possible alternative. They list memory requirements and noisy behavioural data as reasons to avoid it. These are indeed potential issues, but whether they matter in practice should be established. I think the authors should compare their approach to n-gram counting for two reasons: n-gram counting has the advantage of simplicity/intelligibility and it has previously been shown to work in identifying behaviour sequences that distinguish free roaming from chemotaxis (Schwarz et al. listed above). To what extent would over-represented n-grams have identified the aversive sequence bouts they describe or the thermalling strategy of the glider?

The authors also contrast their approach with compression algorithms, stating that they "do not necessarily lead to meaningful motifs; for example, the two-symbol word ab could be identified as a motif simply because a and b occur often, even if a and b occur next to each other purely by chance." This might be true in principle, but it's not clear whether it occurs in practice without a more direct comparison between methods (for example, if the sequence ab is useful in compression and is found to be used more frequently in a fish doing aversive chemotaxis compared to controls it's difficult to argue it's meaningless). Another reason to consider a direct comparison is that compression has a distinct advantage over the current approach in computational complexity. The authors discuss the challenge of scaling their approach to larger datasets. Several algorithms in the Neville-Manning and Witten compression paper the authors cite have linear-time implementations which makes them easy to scale to large behavioural datasets. Knowing which approach to choose for large data sets would be easier if there were a better analysis of the actual advantages and disadvantages of BASS compared to compression.

In summary, the authors emphasise the advantages of their method (lines 58-61) and the disadvantages of other methods but don't do a comparison on actual data. Of their listed advantages, 1 and 3 are shared by n-gram counting and possibly 2, although it would depend on the size and nature of the data. Compression algorithms share all three advantages and have better scaling properties than BASS.

I would recommend the following:

1) (at minimum) Include a more complete discussion of the advantages of disadvantages of BASS and other methods.

2) (probably worth doing) Include a comparison between BASS and n-gram counting.

3) (possibly worth doing) Include a comparison between BASS and an HHMM.

4) (possibly worth doing) Include a comparison between BASS and one of the compression algorithms.

Reviewer #2: In this paper, the authors propose a novel method for analyzing locomotion data under various environmental conditions without accessing stimulus information; they extract sequences (motifs) that rarely occur in the noisy behavioural data obtained from freely-moving animals. They investigate a lexical, hierarchical model of behaviour whose two layers map these motif templates to a latent state space which then underlies the noisy observed action outputs. Further they developed an unsupervised inference algorithm called BASS to identify such motif templates in a given dataset, find their corresponding probabilities and determine how they may underpin the observed behaviour. Finally the authors confront their generative model and inference method using synthetic data, experimental observations of zerbafish (normal and novel chemotaxis assay) and simulated data of soaring gliders. In these sections they also quantify the non-markovian nature of their derived sequences. Additionally, in case of the zebrafish they comment on the implications of their differing motif dictionaries.

The paper is very interesting and presents new innovative and interesting results. Therefore, to my opinion, the manuscript clearly deserves publication in PLoS Computational Biology. However, I have a few detailed remarks, which I list below.

1. It is not clear form the outset what conserved means. My understanding is this means behaviour conserved across different environments (such as circle and square swim areas for the zebrafish) and across genetic variations. Could the authors please clarify this in the introduction?

2. Please check that $p_d$ is referred to as probability of deletion (not insertion) in the manuscript (for eg. Caption of fig. 2, line 254).

3. In fig. 2 based on the text I am inferring that 2(e) also uses same parameters as (b) and (c). Please clarify that this is the case since the parameter values differ in the different subplots. Additionally, I would interested in seeing a similar set of subplots for $epsilon_p \\neq 0$ as a benchmark for future experimental analysis that is presented in the paper.

4. To assist the reader, I would suggest the authors to add a sentence after 211 / in the figure caption stating why optimality is not 100% but ~ 90%.

5. The authors often state that the motifs are rare but then also include that motifs in the case of the moving zebrafish covered 78% of all bouts per fish. I believe it would help clarify the seeming dichotomy if the authors could include a column in tables 1 and 2 stating the fraction of a dataset covered by a motif.

6. Table S1 change 50 to 25.

**Have all data underlying the figures and results presented in the manuscript been provided?**

Reviewer #1: Yes

PLOS authors have the option to publish the peer review history of their article (what does this mean?). If published, this will include your full peer review and any attached files.

Reviewer #1: No

Reviewer #2: No

**Have the authors made all data and (if applicable) computational code underlying the findings in their manuscript fully available?**

Reviewer #2: Yes
---

## [Decision Letter · Decision Letter 1]

16 Nov 2021

Dear Dr. Reddy,

We are pleased to inform you that your manuscript 'A lexical approach for identifying behavioral action sequences' has been provisionally accepted for publication in PLOS Computational Biology.

Best regards,

Aldo A Faisal

Associate Editor

PLOS Computational Biology

Stefano Allesina

Deputy Editor

PLOS Computational Biology

Reviewer's Responses to Questions

**Comments to the Authors:**

Reviewer #1: The authors have addressed my concerns.

Reviewer #2: The authors have correctly answered all of my comments, and for me the readability of the paper has since improved.

**Have the authors made all data and (if applicable) computational code underlying the findings in their manuscript fully available?**

Reviewer #1: None

Reviewer #2: Yes

PLOS authors have the option to publish the peer review history of their article (what does this mean?). If published, this will include your full peer review and any attached files.

Reviewer #1: No

Reviewer #2: No

---

## [Editor Report · Acceptance letter]

17 Dec 2021

PCOMPBIOL-D-20-01469R1 

A lexical approach for identifying behavioural action sequences

Dear Dr Reddy,

I am pleased to inform you that your manuscript has been formally accepted for publication in PLOS Computational Biology. Your manuscript is now with our production department and you will be notified of the publication date in due course.

With kind regards,

Livia Horvath
